# LET'S DO THE TIME-WARP-ATTEND: LEARNING TOPOLOGICAL INVARIANTS OF DYNAMICAL SYSTEMS

**Noa Moriel**[*]
The Hebrew University
noa.moriel@mail.huji.ac.il

**Matthew Ricci**[*]
The Hebrew University
matthew.ricci@mail.huji.ac.il

**Mor Nitzan**[†]
The Hebrew University
mor.nitzan@mail.huji.ac.il

## ABSTRACT

Dynamical systems across the sciences, from electrical circuits to ecological networks, undergo qualitative and often catastrophic changes in behavior, called bifurcations, when their underlying parameters cross a threshold. Existing methods predict oncoming catastrophes in individual systems but are primarily time-series-based and struggle both to categorize qualitative dynamical regimes across diverse systems and to generalize to real data. To address this challenge, we propose a data-driven, physically-informed deep-learning framework for classifying dynamical regimes and characterizing bifurcation boundaries based on the extraction of topologically invariant features. We focus on the paradigmatic case of the supercritical Hopf bifurcation, which is used to model periodic dynamics across a wide range of applications. Our convolutional attention method is trained with data augmentations that encourage the learning of topological invariants which can be used to detect bifurcation boundaries in unseen systems and to design models of biological systems like oscillatory gene regulatory networks. We further demonstrate our method's use in analyzing real data by recovering distinct proliferation and differentiation dynamics along pancreatic endocrinogenesis trajectory in gene expression space based on single-cell data. Our method provides valuable insights into the qualitative, long-term behavior of a wide range of dynamical systems, and can detect bifurcations or catastrophic transitions in large-scale physical and biological systems.

**Keywords:** dynamical systems, bifurcations, topological invariance, Hopf bifurcation, physics-informed machine learning, augmentation, single-cell RNA-sequencing.

## 1 INTRODUCTION

Gradually changing the parameters of a dynamical system will typically result in innocuous, quantitative alterations to the system's behavior. However, in exceptional cases, parameter perturbations will result in a qualitative, potentially catastrophic change in the system's dynamical behavior called a *bifurcation*. Bifurcations may lead to extreme consequences, as when a beam buckles and breaks under increasing weight (Strogatz, 1994), when disastrous oscillations presage the collapse of poorly designed bridges, like the infamous Tacoma Narrows Bridge (Billah & Scanlan, 1991; Deng et al., 2016), or when an airplane exceeds a speed threshold, leading to dangerous self-excited oscillations in its tail (Weisshaar, 2012; Dimitriadis, 2017). In biological systems, bifurcations marking the transition away from oscillatory dynamics on cellular or organ-level scale, such as those regulating

---

[*]Equal contribution.
[†]Corresponding author.

blood pumping and circadian rhythms, can have dire health consequences (Xiong & Garfinkel, 2023; Qu et al., 2014).

Understanding bifurcations which transition between dynamical regimes, be they in physics, engineering, or biology, is a subject of intense research in the study of nonlinear systems (Kuznetsov, 1998; Karniadakis et al., 2021; Ghadami & Epureanu, 2022). One of the central challenges across all applications is that distinct dynamical regimes can be qualitatively identical (e.g. the emergence of oscillations) while being quantitatively very different (e.g. the amplitude or frequency of those oscillations); smoothly warping the state space has no qualitative effect on the dynamics. Put another way, a detector for different dynamical regimes and the transitions between them must be selective for *topological* structure of underlying dynamics but invariant to the *geometric* nuisances of those dynamics. Predicting the onset of such bifurcations often requires the explicit use of governing equations, which must be hardwired or inferred. Hence, the analysis of real data, where such equations are often unknown or difficult to define, is particularly challenging (Kuznetsov, 1998). Applications to real data either use hardwired (Wissel, 1984; Wiesenfeld, 1985; Scheffer et al., 2009) or learned (Bury et al., 2023) time series features, but no approach, to our knowledge, explicitly addresses the selectivity-invariance trade-off inherent to the topological nature of the problem. Additionally, where vector field data is readily available and time-series tracing is not, e.g., in capturing gene expression of cells in high-throughput single-cell experiments, it remains unclear how to extract topological information either from such noisy and sparse vector fields directly or from their integration.

We address this challenge by learning a universal dictionary of dynamical features which is explicitly encouraged to be invariant to geometric nuisances but selective for the topological features which define a particular type of bifurcation (Fig. 1). This is achieved primarily by the use of *topological data augmentation*: each sample of our training data is a velocity field which comes bundled with numerous deformed but identically labeled versions. The features extracted from these augmented samples are then passed to a supervised classifier which must contend with the substantial geometric nuisances induced by the augmentation scheme. We demonstrate our method for the ubiquitous supercritical Hopf bifurcation whereby a stable fixed point (or "equilibrium") loses stability and transitions into a regime exhibiting a stable limit cycle. We train on synthetic, augmented data generated from a simple oscillator prototype. Despite being only trained on data from one simple equation, our architecture can generalize to radically different systems, such as the nonlinear Selkov glycolysis oscillator and the Belousov-Zhabotinsky model of chemical oscillations.

Our approach is geared towards realistic, high-dimensional, noisy, and complex data. For instance, we apply our technique to the repressilator, a gene regulatory network which can generate oscillatory gene expression, and infer the probability for oscillatory behavior under conditions of varying gene transcription and degradation rates. Finally, we apply our method to the analysis of complex, large-scale biological data. By extracting features from single-cell gene expression velocity data during the differentiation of pancreatic cells, our method can distinguish between proliferation and differentiation dynamics with high fidelity. Our results indicate that our approach can be applied to a broad range of dynamical systems, providing valuable insights into their dynamic characteristics and bifurcation behavior [1].

The main contributions of this paper are:

1. A novel approach to topologically invariant feature learning in dynamical systems. The approach uses warped vector field data as augmentations to generalize across systems exhibiting similar dynamics in terms of their topological representation in phase space, allowing for a more selective and invariant understanding of bifurcations in diverse real-world systems.

2. Invariant classification of complex and nonlinear synthetic data, including models of diverse chemical and biological systems, achieved by leveraging knowledge of a single prototypical system.

3. The identification of bifurcation boundaries using classifier robustness.

4. The recovery of distinct proliferation and differentiation dynamics along the pancreatic differentiation trajectory from single-cell gene expression data, providing a practical application of our method to the analysis of biological systems.

---

[1]Code available at: https://github.com/nitzanlab/time-warp-attend

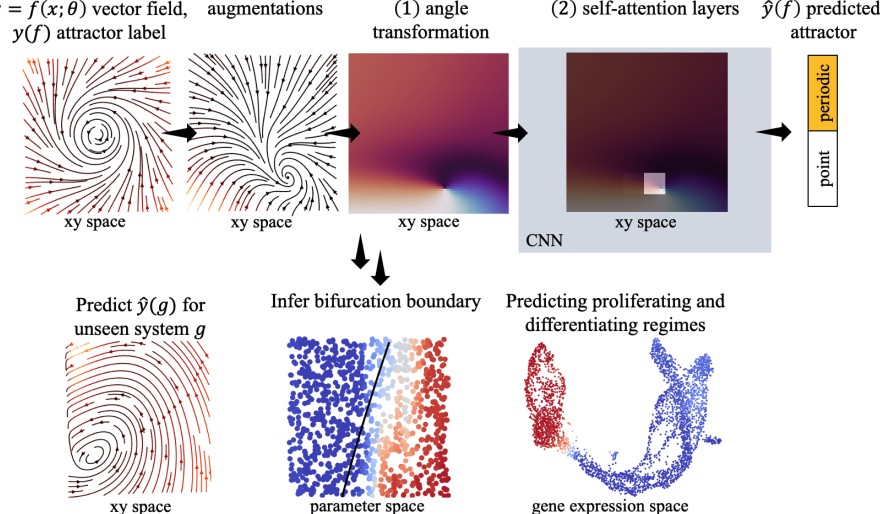

**Figure 1:** *Overview.* Our framework classifies long-term dynamical behavior of point vs periodic attractor across systems and regimes by leveraging topologically-invariant augmentations of a single prototypical system. To encourage generalization, we (1) abstract system-specific vector magnitudes and use angular representations instead, and (2) deploy attention layers which focus on learning essential dynamical cues. Using this framework, we are able to classify and recover the bifurcation boundaries of complex, unseen systems. We apply this method to the problem of recovering cell-cycle scores which distinguish between proliferating and differentiating cell populations. All vector fields are plotted as streamlines.

## 2 RELATED WORK

**Classical methods for detecting limit cycles.** Solving non-linear differential equations is in general difficult, and it has been recognized since the time of Poincaré that reasoning about dynamical systems' characteristics can be made easier by qualitatively delineating solutions according to their long-term behaviors. For autonomous (i.e., time-invariant) systems in two dimensions, if the dynamics do not diverge, these behaviors are either fixed points or limit cycles, and transitions between these dynamical regimes as a result of changing parameters are called bifurcations. More explicitly, a bifurcation in these cases corresponds to a change in the topological structure of a phase portrait as a parameter of the dynamical system is varied.

Full characterization of the location and number of limit cycles in two-dimensional, polynomial systems is a challenging open problem in dynamical systems theory (Hilbert, 2019; Smale, 2000). Early theoretical results guaranteeing the existence or absence of a limit cycle in two-dimensional autonomous systems are the theorems of Poincaré-Bendixson and Bendixson-Dulac, respectively. Further guarantees are possible, for example, when the system can be reduced to the form of a so-called Liénard equation. None of these results, however, is constructive or easy to use in practice. Center manifold theory (Carr, 2012), for its part, analyzes dynamics on representative subspaces where systems reduce to normal forms having known bifurcation behaviors, though this method requires knowledge of the underlying governing equations. The same limitation applies to Conley-Morse theory, which represents dynamics as a graph whose nodes correspond to long-term behaviors (Conley, 1978). Importantly, none of these approaches is easily adapted to real data, where equations are in general unknown and system measurements can be sparse, incomplete and noisy.

**Data-driven methods for detecting limit cycles.** Data-driven methods fall into non-topological and topological classes. In the former class, several approaches use compressed sensing of trajectory data to reduce noise, followed by identification of system parameters and associated bifurcation diagrams (Wang et al., 2011; 2016), though this approach requires inferring governing equations for each new sample. Other work sought to reduce vector field measurements to a Conley-Morse graph (Chen et al., 2007b;a) for which a database over parameter regimes can be constructed via combinatorial dynamics exploration and graph equivalence comparisons (Arai et al., 2009). Yet other approaches

used estimates of Lyapunov exponents to delineate between periodic and stationary systems, although, as we show below, this approach can be sensitive to noise (Stoker, 1950; Zhang et al., 2009).

On the topological front, there is a growing body of work using persistent homology to detect periodicity in time series data (Ravishanker & Chen, 2021). For example, Perea & Harer (2015) extracted persistence barcodes from delay embeddings of time series data to measure periodicity, although this process is computationally expensive and there is no standard approach to classifying barcodes. Persistent homology approaches often assume that the underlying bifurcation parameter is known (Tymochko et al., 2020), which greatly simplifies the problem at the cost of realism. To our knowledge, there is no data-driven method which is efficient (not requiring expensive barcode calculations for each sample), general (applicable to novel systems without retraining) and robust (not sensitive to noise).

**Generalization across dynamical systems.** Comparatively few approaches exist for the simultaneous analysis of diverse dynamical systems and their bifurcation behavior. Work by Brown & Sethna (2003); Transtrum & Qiu (2014) and Quinn et al. (2021) mapped families of dynamical systems to a manifold whose geometry encodes different qualitative behaviors, but these methods have no straightforward way of identifying bifurcations between these behaviors. More recent work from Ricci et al. (2022) uses a data-driven approach called `phase2vec` to learn manifolds of dynamical systems, leading to dynamical embeddings which encode qualitative properties of the underlying systems, though it is not tailored for the detection of distinct dynamical regimes and their associated bifurcations per se.

**Diffeomorphic augmentations for generalization.** Diffeomorphic transformations which preserve topology in the context of dynamical systems have been used principally in computer vision. In order to train a classifier of a limited image set of biological entities such as diatoms, Vallez et al. (2022) used diffeomorphic augmentations to expand its train data, however, since the labels of these data were not necessarily preserved by these transformations, a manual post-filtering was required. In a second set of works concerning dynamical systems specifically, Khadivar et al. (2020) and Bevanda et al. (2022) fit a diffeomorphic transformation into closed or linearized forms, and work by Bramburger et al. (2021) used an autoencoder approach to reduce a complex dynamical system to a topologically conjugate (i.e., dynamically equivalent) form of discrete-time mapping. Here, we use diffeomorphisms to curate a set of consistently labeled dynamical augmentations.

## 3 METHODOLOGY

**Topological equivalence in dynamical systems.** Let $\dot{x} = f(x; \theta)$ for $x \in \mathbb{R}^2$ be a parameterized dynamical system with parameter vector $\theta \in \Theta$. A solution to this equation, $F(x_0, t)$, with an initial condition, $x(t = 0) = x_0$, is called a trajectory through $x_0$. A standard approach to understanding dynamical systems (Kuznetsov, 1998) is to characterize them by the topology of all of their trajectories, construed as functions of $x$ and $t$ or just $x$. For example, do trajectories extend indefinitely, stop at a point (a fixed point) or loop back on themselves (a limit cycle)? More precisely, we say that $f$ has a fixed point at $x^*$ and $\theta = \theta^*$ if $f(x^*; \theta^*) = 0$. We say that $f$ has a limit cycle at $\theta^*$ if there exists a trajectory of $f$ which is closed and isolated (neighboring trajectories are not closed). Importantly, these properties are preserved when the state space is subjected to a topology-preserving map, making them topological invariants of the dynamical systems. However, these properties may change as $\theta$ changes; i.e. when the system undergoes a bifurcation. For example, in a system that undergoes a supercritical Hopf bifurcation, there exists a parameter $\theta$ such that below a certain threshold of $\theta$ the system exhibits a stable fixed point, and above it, the fixed point loses its stability and a stable limit cycle emerges. See Appendix 7.1 for definitions.

**Dynamical equivalence and topological augmentations.** In what follows, we will consider autonomous systems with either a single stable fixed point (i.e., equilibrium) or a stable limit cycle with an unstable fixed point in its interior. We term these two regimes as *dynamical classes*, denoted $y(f)$, and we denote the fixed point and limit cycle classes considered here the "pre-Hopf" and "post-Hopf" classes, respectively. As discussed above, these classes are invariant to topology-preserving deformations of the state space. Here, we will use diffeomorphisms, $h \in H$ which induce an equivalence relation on dynamical systems in a standard and well-studied way: we have $y(f) = y(f \circ h)$ for all $h \in H$ so that a given $H$ defines an equivalence relation which partitions the

space of dynamical systems into its dynamical classes. We refer to two parameterized systems, $f, g$ such that $g = f \circ h$ as being *dynamically equivalent* [2].

In this light, classification becomes a problem of learning representations of dynamical systems which are invariant to all $h \in H$ but selective for the underlying dynamical class. To that end, we learn features on numerous, transformed versions of a prototypical system, a simple oscillator $f$ defined for a point at radius $r$ and angle $\phi$ with parameters $\theta = (a, \omega)$:

$$\dot{r} = r(a - r^2)$$
$$\dot{\phi} = \omega,$$

which has a sparse functional form and undergoes a supercritical Hopf bifurcation for any $\omega \neq 0$ at $\theta^* = (0, \omega)$. We validate these learned features by measuring how well they encode the dynamical class and bifurcation boundary of testing data, which, though superficially very different from the training data, having different functional forms and parameter regimes, are nevertheless in the same equivalence classes induced by $H$.

Our training data is constructed by first acquiring planar vector fields, $\{f(x, \theta) \in \mathbb{R}^2\}$ evaluated at all points, $x$, on a fixed, $64 \times 64$ lattice. Parameters, $\theta$, are sampled uniformly on $a \in [-0.5, 0.5], \omega \in [-1, 1]$, and each system is assigned a label according to whether it is in the pre- or post-Hopf regime. Then, the dataset is augmented with diffeomorphisms generated using monotonic rational-quadratic splines. In particular, we use the spline construction from Durkan et al. (2019), where instead of training the neural spline flows proposed in that work, we use its random initialization which generates a diverse set of diffeomorphic augmentations. For details on data augmentation, see Appendix 7.2.1.

**Convolutional attention training.** We construe our full, augmented data set as a collection of labeled arrays in $\mathbb{R}^{2 \times 64 \times 64}$ which we use to train a convolutional network. Our architecture is adapted from the discriminator of self-attention generative adversarial network (SAGAN) of Zhang et al. (2019), coupled with a multi-layered perception (MLP). For standardization of the data, we reduce each vector field to only its angular components, ignoring magnitudes which carry little information about cyclic behavior. The feature extraction module consists of four convolutional blocks with self-attention mechanism into the last two convolutional layers. Our reasoning for incorporating self-attention is that only a few dynamical keypoints in the vector field (i.e., the fixed point or the limit cycle) carry information about class label, and the rest of the array can be (nearly) ignored. Together, the data warping and attentional components of our pipeline comprise our eponymous method. Features from this network are passed to a multilayer perceptron classifier which is trained with a supervised, cross-entropy loss. Because of our augmentation scheme, this classifier must contend with data which, though radically different in appearance, belong to the same dynamical class, encouraging the classifier to learn the true dynamical invariances underlying the Hopf bifurcation. Training and architecture details are described in Appendix 7.2.2.

## 4 RESULTS

### 4.1 TRANSFERRING KNOWLEDGE OF DYNAMICAL CLASSES FROM SIMPLE PROTOTYPES TO COMPLEX TEST DATA

We evaluate our model and its baseline competitors on a diverse, synthetic dataset of noisy (zero-mean gaussian), real-world systems drawn from across the sciences. Test datasets include the simple harmonic oscillator ("SO"), its warped variant ("Augmented SO"), a system undergoing a supercritical Hopf bifurcation as described in (Strogatz, 1994)("Supercritical Hopf"), examples based on Liénard equations ("Liénard Polynomial" and "Liénard Sigmoid"), and systems used to model oscillatory phenomena in nature and engineering applications, namely the van der Pol system, used for modeling electrical circuits in early radios ("Van der Pol"), the Belousov-Zhabotinsky model, a model of chemical oscillations ("BZ reaction"), and the Selkov oscillator, used in modeling glycolitic cycles ("Selkov"), see Appendix 7.3 and Strogatz (1994).

Our baselines comprise: (1) "Critical Points", a heuristic based on critical points identification as derived from Helman & Hesselink (1989; 1991) algorithm, (2) "Lyapunov" which distinguishes pre-

---

[2]We choose the neutral term "dynamical equivalence" to distinguish our notion from the similar notions of topological conjugacy and topological equivalence (Kuznetsov, 1998). See Appendix 7.1.2 for details.

and post-Hopf classes according to the value of the Lyapunov exponent of a time series integrated from the vector field, (3) "Phase2vec", vector field embeddings from Ricci et al. (2022), a state-of-the-art deep-learning method for dynamical systems embedding, and (4) "Autoencoder", latent features of a convolutional vector field autoencoder, see Appendix 7.4. As with our method, for all baselines except Critical Points, classifications are produced by a linear classifier trained on features from augmented simple harmonic oscillator data (Augmented SO) in order to discriminate between point and cyclic attractors. The accuracy average and standard deviation for our model and the trained baselines are computed over 50 training runs.

We find that our framework generalizes across a wide range of dynamical systems and classifies distinct dynamical regimes in noisy (zero-mean gaussian) conditions better on average than baseline methods (see Table 1, Table A4, and examples in Fig. A8). Critical Points achieves nearly perfect classification on noiseless data but is easily disrupted by noise (e.g., the perfect classification of Supercritical Hopf drops to 56% with Gaussian noise $\sigma = 0.1$). Likewise, Lyapunov suffers a substantial decrease in accuracy for classical oscillators (e.g., 89% accuracy of Supercritical Hopf reduces to 70% accuracy with Gaussian noise $\sigma = 0.1$). Both with and without noise, Phase2vec extends mainly to SO and Supercritical Hopf. Autoencoder obtains competitive results but exhibits little generalization to Liénard Polynomial system, and essentially no generalization to Selḱov system. We also test resilience to noise in the Augmented SO data (see Table 2). Despite the substantial distortion of the vector fields caused by noise, the accuracies of our model and that of autoencoder representation only gradually decrease, unlike the sharp decrease in accuracy for the other baselines.

**Table 1:** Test accuracy with added Gaussian noise of $\sigma = 0.1$.

|  | SO | Aug. SO | Supercritical Hopf | Liénard Poly | Liénard Sigmoid | Van der Pol | BZ Reaction | Selḱov | Average |
|---|---|---|---|---|---|---|---|---|---|
| Our Model | **0.98±0.01** | **0.93±0.01** | **0.99±0.01** | **0.86±0.13** | **0.92±0.07** | 0.83±0.15 | 0.82±0.11 | **0.65±0.03** | **0.87** |
| Critical Points | 0.54 | 0.56 | 0.56 | 0.70 | 0.49 | 0.56 | **0.84** | 0.51 | 0.60 |
| Lyapunov | 0.64 | 0.60 | 0.71 | 0.85 | 0.67 | **0.87** | 0.83 | 0.57 | 0.72 |
| Phase2vec | 0.73±0.08 | 0.71±0.04 | 0.88±0.03 | 0.49±0.06 | 0.48±0.0 | 0.49±0.04 | 0.5±0.0 | 0.49±0.0 | 0.6 |
| Autoencoder | 0.95±0.03 | 0.87±0.01 | 0.97±0.01 | 0.63±0.16 | 0.88±0.09 | 0.81±0.16 | 0.82±0.13 | 0.49±0.02 | 0.8 |

**Table 2:** Evaluation of Augmented SO with added Gaussian noise of scale $\sigma$.

| $\sigma$ | 0 | 0.10 | 0.20 | 0.30 | 0.40 | 0.50 | 0.60 | 0.70 | 0.80 | 0.90 | 1.00 |
|---|---|---|---|---|---|---|---|---|---|---|---|
| Our Model | **0.93±0.01** | **0.93±0.01** | **0.93±0.01** | **0.91±0.02** | **0.86±0.04** | **0.78±0.06** | **0.7±0.06** | 0.64±0.04 | 0.6±0.04 | 0.57±0.03 | 0.54±0.02 |
| Critical Points | 0.93 | 0.59 | 0.53 | 0.507 | 0.51 | 0.51 | 0.51 | 0.51 | 0.51 | 0.50 | 0.50 |
| Lyapunov | 0.71 | 0.60 | 0.52 | 0.50 | 0.50 | 0.50 | 0.50 | 0.50 | 0.50 | 0.50 | 0.50 |
| Phase2vec | 0.87±0.03 | 0.7±0.05 | 0.53±0.01 | 0.51±0.01 | 0.5±0.0 | 0.5±0.0 | 0.5±0.0 | 0.5±0.0 | 0.5±0.0 | 0.5±0.0 | 0.5±0.0 |
| Autoencoder | 0.86±0.01 | 0.87±0.01 | 0.84±0.02 | 0.8±0.04 | 0.76±0.05 | 0.72±0.05 | 0.69±0.06 | **0.68±0.06** | **0.65±0.06** | **0.64±0.05** | **0.62±0.05** |

In Appendix 7.4, we report the results of another baseline, "Parameters" which trains a linear classifier based on a flattened vector of coefficients of a degree three polynomial fit to each vector field with least squares, a common method for bifurcation analysis, although this resulted in a poor average accuracy of 53%.

Furthermore, we find that all modules in our pipeline (topological augmentation, angular representation of data, self-attention layers) contribute, on average, to accuracy in complex systems, see Appendix 7.5. Concretely, the average classification accuracy of the framework, trained on data without augmentations, using vector representation, lacking attention layers, or without any of these, is 78%, 82%, 77%, 68%, respectively, and notably, except for the ablation of attention layers, other ablated models do not generalize to the complex BZ reaction and Selḱov systems. We also found that topological augmentations are particularly necessary when generalizing to systems with a fixed point far from the center of the grid. For example, "centered" systems such as Liénard Sigmoid and Liénard Polynomial, are well-characterized without augmentations, reaching 0.96 and 0.94 accuracy respectively, whereas systems that are not centered, e.g. BZ reaction whose fixed point is close to the edge of the grid, require augmentations for generalization.

We also evaluate the robustness of our method as a function of (1) the training dataset size, where we find good generalization even on relatively small data sets (0.84 accuracy after training on $\sim 2k$ vector fields), (2) vector field resolution, where performance is stable when doubling the vector resolution (0.86 accuracy), and (3) architectural and optimization hyperparameters (learning rate, training epochs, etc.), obtaining almost consistently an overall accuracy of between 0.85 and 0.88. See Appendix 7.5 and Table A4 for details.

## 4.2 RECOVERING BIFURCATION BOUNDARIES

Classifier confidence is expected to decrease for systems near the bifurcation boundary since limit cycles in this regime shrink below the vector field resolution, making them indistinguishable from fixed points. This effect can be used to pinpoint the bifurcation boundary in parameter space. For example, when we plot each system's average prediction as a function of its bifurcation parameter (for Augmented SO, the parameter values are chosen as the original $a, \omega$), we find an inverse relation between the parametric distance from the bifurcation boundary and classifier confidence (see Fig. 2, Fig. A11 and Fig. A12 for bifurcation diagrams of alternative baselines). An exception is the upper boundary of the Selḱov system where instances of high $b$ value are misclassified as exhibiting cyclic dynamics, and hence, the upper bifurcation boundary in the $a - b$ parameter plane is undetected (Fig. 2, right plot). We explain the systematic misclassification as a result of damping which makes trajectories move in slow, tight spirals towards the fixed point attractor in a manner numerically indistinguishable from limit cycles (Fig. A7). These overall suggest that our method can be geared towards recovering the bifurcation boundaries of unseen systems.

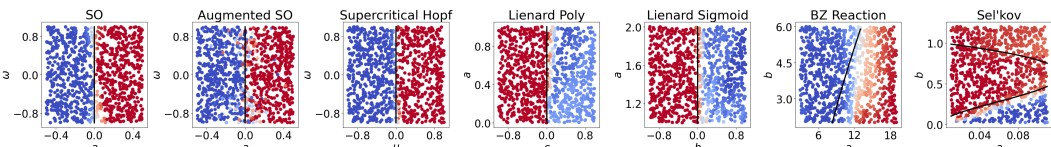

**Figure 2:** *Inference of bifurcation boundaries.* Each system is plotted at coordinates of its simulated parameter values and colored according to its average cycle prediction across 50 training re-runs. The true parametric bifurcation boundary is plotted in black. Colors are scaled between a perfect cycle classification (red) and perfect point classification (blue) with white at evenly split classification.

## 4.3 PREDICTING OSCILLATORY REGULATION OF THE REPRESSILATOR SYSTEM

In order to demonstrate the viability of our method for higher-dimensional systems under realistic noise conditions, we apply our topological approach to a computational model in a well-studied biological setting, the so-called repressilator model (Elowitz & Leibler, 2000). The repressilator system is a synthetic, genetic regulatory network that was introduced by Elowitz and Liebler into bacteria to induce oscillatory regulation of gene expression (Fig. 3). The system is modeled by the mRNA and protein counts of three genes (TetR, LacI and $\lambda$ phage cI) that inhibit each other in a cyclic manner (see Appendix 7.8 for governing equations). Theoretically and experimentally, the existence and characteristics of oscillations have been shown to depend on numerous factors like the rates of transcription, translation, and degradation, as well as intrinsic and extrinsic noise (Verdugo, 2018; Potvin-Trottier et al., 2016). Here we test our method in classifying a regime of the repressilator which has been shown to undergo a Hopf bifurcation, though, unlike the previous complex simulated systems, it spans six dimensions and is constructed from noisy cellular time-series.

To generate 2D vector field data, we consider trajectories of cellular gene expression measured by protein levels of TetR and LacI (Fig. 3, middle). Different instances of simulations share the initial condition but vary by the rate of transcription, $\alpha$, and by the ratio of protein and mRNA degradation rates, $\beta$. We sample 100 cell states along the temporal trajectory and add Gaussian noise of $\sigma = 0.5$ to the gene expression states (to protein and mRNA levels). For each set of measurements, we interpolate an input 64x64 vector field, based on their velocities across TetR-LacI protein phase space, see examples at Fig. A14 and Appendix 7.8 for further details. We predict for each vector field its dynamical class, achieving 87% classification accuracy and proximate bifurcation boundary to the analytically-derived boundary (Fig. 3, right (Verdugo, 2018)). In Fig. A13 we show prototypes of trajectories across the $\alpha - \beta$ parameter space, where we see that where we fail to classify a point attractor, trajectories of cells loop closely in the TetR-LacI protein plane and generate data samples that highly resemble trajectories with a periodic attractor rather than with a point attractor.

## 4.4 DISTINGUISHING DYNAMICAL REGIMES IN SINGLE-CELL DATA OF THE PANCREAS

Last, we leverage our method to characterize distinct dynamical regimes corresponding to different biological processes in real data. Specifically, we focus on distinguishing proliferation vs. differ-

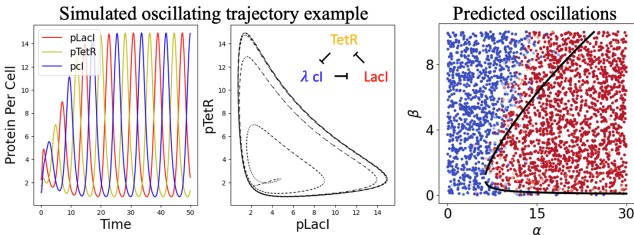

**Figure 3:** *Oscillations in a repressilator system.* Example of oscillations in the repressilator system viewed as a time-series *(left)* or projected onto TetR-LacI protein plain *(middle)*. *(right)* Predicted oscillatory behavior by our model output upon varying transcription rate without repression ($\alpha$) and the ratio of protein and mRNA degradation rates ($\beta$). The theoretical boundary is marked in black. See details in Appendix 7.8.

entiation dynamics in single-cell RNA-sequencing data. A population of proliferating, precursor cells expresses genes associated with DNA duplication in a cyclic fashion (Tirosh et al., 2016; Schwabe et al., 2020; Karin et al., 2023). Cell differentiation, on the other hand, and similarly to the Waddington landscape abstraction (Waddington, 2014), is often characterized by cellular flow in gene expression space towards a localized, differentiated state (Sáez et al., 2022). We show that our model can detect the qualitative change in the flow of cells in gene expression space as they mature and differentiate out of their precursor cell state, moving from a dynamical regime characterized by a periodic behavior to that characterized by fixed points.

We focus on the process of pancreas endocrinogenesis where Ductal and Ngn3+ precursor cells rapidly proliferate to give rise to functional cell types Alpha, Beta, Delta and Epsilon (Bastidas-Ponce et al., 2019); In that study, a single-cell RNA-sequencing dataset was collected where the expression of over 2,000 genes are measured per cell, for a total of 3,600 cells (Fig. 4, left). While the RNA expression profile, providing a gene expression vector per cell, provides for each cell its location within gene expression space, RNA velocity (Bergen et al., 2021; Gorin et al., 2022) can be used to compute the effective velocity of each cell (based on differences in spliced and unspliced RNA counts). We use scVelo (Bergen et al., 2020) to infer cellular velocities in the high-dimensional gene expression space and to project them to a 2D UMAP representation. Simultaneously, we also compute for each cell a 'cell-cycle score', which captures the extent of expression of cell-cycle genes (in phase S and in grouped G2 and M phases, Fig. 4, middle; see Appendix 7.9, Bergen et al. (2020); Tirosh et al. (2016)) . For each set of cells, we interpolate an input 64x64 vector field, based on their RNA velocities, over the 2D UMAP representation.

To generate multiple frames from the trajectory, we randomly split cells of each cell type into groups of at least 50 cells resulting in 69 (sparse, noisy) vector fields with corresponding cell-cycle score (Fig. 4, middle, Appendix 7.9). We then used our model (still trained solely on simulated augmented simple oscillator data) to characterize the dynamic regimes of the cellular behavior. Our model can successfully identify proliferative, cyclic behavior within the single-cell data, as reflected by 94% accuracy in predicting a cyclic behavior matching high cell cycle scores (Fig. 4, right), substantially exceeding the prediction accuracy reached by alternative baselines (Critical Points: 54%, Lyapunov: 49%, Phase2vec: 30%, Autoencoder: 30%). As opposed to the cell-cycle score, our model does not use any prior knowledge regarding cell-cycle related genes, and is only based on the structural features of the cellular vector fields, and therefore can also be employed when such prior knowledge is unavailable.

## 5 LIMITATIONS

Our framework learns to be invariant to geometric (as opposed to topological) properties of systems through topological data augmentation and convolutional attention, achieving competitive results in bifurcation detection across real-world systems. Nevertheless, we note here several important limitations of our method in its current form. The method is currently confined to the important but circumscribed setting of two-dimensional dynamics. Scaling up to much higher dimensions in these contexts presents technical challenges like the exponential growth of input size with dimension, as well as theoretical challenges such as the lack of structural stability in higher dimensions and the introduction of chaotic regimes. Since inherently low-dimensional dynamics often underlie high-

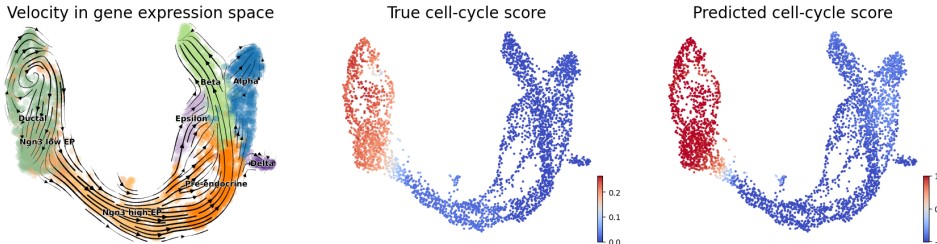

**Figure 4:** *Proliferation-to-differentiation transition in pancreas development. (left)* Gene expression velocity of cells undergoing pancreas endocrinogenesis in UMAP space. Colors distinguish between different cell types. *(middle)* Cell-cycle score based on the fraction of cells with high S-phase gene expression (with S-score $> 0.4$). *(right)* Predicted cell-cycle score computed as the average classification prediction across 50 training re-runs. See details in Appendix 7.9.

dimensional measurements, especially in biology (Xiong & Garfinkel, 2023; Khona & Fiete, 2022), high dimensions can be mitigated by hardwired, or optimized dimensionality reduction methods as we deploy for the six-dimensional repressilator and for analyzing over 2,000 gene expression (dimensional) dynamics of the pancreas single-cell dataset. Additionally, we used a supervised, multi-class setup so that the classifier can predict that both a limit cycle and a fixed point co-exist. Empirically, we find that our method learns that the point and periodic attractors are exclusive, in fact, the learned logits consistently negate each other. For emphasis, we tested our trained model on samples representing a subcritical Hopf bifurcation having a regime with point and periodic attractors whose logits were of intermediate value but still negations, see Appendix 7.7. In the future, we aim to extend to combinatorial annotations of invariant sets, as elaborated in the discussion.

## 6 DISCUSSION

The problem we consider is the data-driven classification of a diverse set of both real and synthetic dynamical systems into topological equivalence classes. This broad scope fundamentally distinguishes our work from most existing methods in bifurcation analysis which deal exclusively with the prediction of oncoming catastrophes from time series data of single, isolated systems. The fact that our method can be simultaneously applied to numerous systems of diverse functional and parametric forms demonstrates that our model has learned an abstract notion of these equivalence classes: it can be applied out-of-the-box to novel scenarios without the need for retraining. The integration of physics-informed data augmentation within the pipeline is crucial for attaining this high-level performance for complex systems that are substantially different from the prototype system underlying the training data. Our model captures visual uncertainty in systems near bifurcation boundaries, allowing for the extraction of an equation-free representation of these transitions. Furthermore, our model demonstrates robustness across samples disrupted by noise, withstands data produced from complex, high-dimensional time-series that is prominent for real-world applications such as the design of synthetic gene regulation, and successfully detects in real single-cell data the qualitative shift in cellular flow topology, corresponding to shifts of biological behaviors, from precursor to differentiated cell states, without relying on gene semantics.

Our work suggests numerous next steps including further translating our prior knowledge of topological properties, like the importance of critical points, into either explicit features or trainable, physics-informed network components (Karniadakis et al., 2021). Another goal is to broaden the scope to encompass a wider range of invariance types, combinatorial comparisons of invariant sets (e.g., via approximation of Conley-Morse graphs), and extensions to stronger notions of equivalence, such as topological conjugacy. To accommodate more versatile invariances and bifurcations, future work could extend our framework to the unsupervised case. For example, given that our data augmentations produce samples which are in the same dynamical equivalence class, a contrastive loss could be used to shape the feature space so that dynamical classes are represented implicitly during training. Or, while challenging, following Skufca & Bollt (2008), a purely unsupervised approach could be used to explicitly construct warpings between systems in order to characterize equivalence classes. Our work demonstrates the potential for scaling up to applications such as identifying dynamical regimes and underlying driving factors across both biological and physical applications.

ACKNOWLEDGMENTS

We thank Yedid Hoshen for early discussions. This work was supported by the Israeli Council for Higher Education Ph.D. fellowship (N.M.), the Center for Interdisciplinary Data Science Research at the Hebrew University of Jerusalem (N.M.), the Zuckerman Postdoctoral Program (M.R.), Azrieli Foundation Early Career Faculty Fellowship, Alon Fellowship, and the European Union (ERC, DecodeSC, 101040660) (M.N.). Views and opinions expressed are however those of the author(s) only and do not necessarily reflect those of the European Union or the European Research Council. Neither the European Union nor the granting authority can be held responsible for them.

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

## 7 APPENDIX

### 7.1 DEFINITIONS

#### 7.1.1 DYNAMICAL SYSTEMS AND INVARIANT SETS

This paper is concerned with dynamical systems which can be identified (potentially implicitly) with two-dimensional systems of ordinary differential equations. For a more general treatment, see Strogatz (1994); Kuznetsov (1998).

Let $z = (x, y) \in \mathbb{R}^2$ and

$$\frac{dz}{dt} = \left( \frac{dx}{dt}, \frac{dy}{dt} \right) = f(z, t) \tag{1}$$
$$z(0) = z_0$$

be an initial value problem for $f$ assumed to be smooth.

**Definition 1** (Continuous-time real dynamical system). *The map $\phi : \mathbb{R}^2 \times [0, T] \to \mathbb{R}^2$ given by*

$$\phi(z_0, T) = z(0) + \int_0^T f(z(t), t) \, dt \tag{2}$$

*is called a (continuous-time real) **dynamical system** or **flow**. In this context, $\mathbb{R}^2$ is called the **phase space**, $z(t)$ is a particular **state** of $\phi$ at $t$ and $f(z(t), t)$ is called the **vector field** associated to $\phi$. For a fixed $z_0$, the map $\phi_{z_0} : \mathbb{R}^2 \to \mathbb{R}^2$ given by a particular solution to Eq. 1 is called a **trajectory** of $\phi$ through $z(0) = z_0$.*

Hereafter, we assume $f(z(t), t) = f(z(t))$, in which case we call $\phi$ **autonomous**. Otherwise, it would be called **non-autonomous**.

The study of dynamical systems is characterized by its focus on long-term behavior of such systems. In our case, two long-term behaviors are of interest. First, if there exists $z^* \in \mathbb{R}^2$ such that $f(z^*) = \mathbf{0}$, then we say $z^*$ is a **fixed point** of $\phi$. In particular, $\phi(z^*, T) = z^*$ for all $T$.

The second long-term behavior of interest in our context is the **limit cycle**.

**Definition 2** (Limit cycle). *Suppose there exist $z_0$ and $T$ such that $\phi_{z_0}(T) = z_0$. Then, the set*

$$\gamma = \{ z \in \mathbb{R}^2 \ : \ \exists \, t \text{ s.t. } \phi_{z_0}(t) = z \} \tag{3}$$

*is a closed, simple curve (simple since Eq. 1 always has a unique solution). If for each $z \in \gamma$ there exists an $\epsilon > 0$ such that the ball $B_\epsilon(z)$ of radius $\epsilon$ around $z$ is intersected by no other closed, simple curve, then we say $\gamma$ is isolated. A trajectory of $\phi$ forming an isolated, simple, closed curve, $\gamma$, is called a **limit cycle**.*

Zero and one-dimensional structures like fixed points and limit cycles are examples of "invariant sets" and serve to distinguish different types of long-term behaviors in dynamical systems. Further distinctions can be made by considering the *stability* of these structures: informally, how the system behaves when it is slightly perturbed away from a fixed point or limit cycle. In the following definition, the set $S$ will denote either a singleton fixed point or the curve in $\mathbb{R}^2$ coinciding with a limit cycle. For any $z$, let

$$d(z, S) = \inf \{ \|z - z'\|_2 \ : \ \forall z' \in S \} \tag{4}$$

**Definition 3** (Asymptotic stability). *Let $S$ be an invariant set as defined above with respect to a given dynamical system, $\phi$. If there exists a $\delta > 0$ such that when $d(z, S) < \delta$ and $d(\phi(z, t), S) \to 0$, then we say $S$ is (asymptotically) **stable**. Otherwise, we say it is (asymptotically) **unstable**.*

Informally, $S$ is stable when small perturbations away from $S$ eventually converge onto $S$.

#### 7.1.2 EQUIVALENCE OF DYNAMICAL SYSTEMS

Fixed points and limit cycles are invariant sets not just because if a solution starts in or enters the set it stays within it, but also because those sets are preserved under the action of certain transformations

to the whole system. Informally, these transformations continuously and smoothly warp the phase space in a way that preserves fixed points, limit cycles and their stability properties. Systems which differ only by such a transformation are considered equivalent. More specifically, recall that a **homeomorphism** is a continuous bijection between two topological spaces with a continuous inverse. A **diffeomorphism**, similarly, is a differentiable bijection with a differentiable inverse. We define several notions of equivalence:

**Definition 4.** *Let $\phi(x,t)$ and $\psi(y,t)$ be dynamical systems (flows) on $\mathbb{R}^2$. Then,*

1. ***Topological conjugacy***. *If there exists a homemorphism, $h : \mathbb{R}^2 \to \mathbb{R}^2$, such that*

$$\phi(h(y),t) = h(\psi(y,t)) \tag{5}$$

*for all $y,t$, then $\phi$ and $\psi$ are called "topologically conjugate."*

2. ***Topological equivalence***. *Let $\mathcal{O}$ denote a trajectory of $\phi$. Suppose there exists a homeomorphism $h : \mathbb{R}^2 \to \mathbb{R}^2$ such that*

$$h(\mathcal{O}(y,\psi)) = \{h \circ \psi(y,t) : t \in \mathbb{R}\} = \{\phi(h(y),t) : t \in \mathbb{R}\} = \mathcal{O}(h(y),\phi) \tag{6}$$

*and for each $y$, there is a $\delta > 0$ such that if $0 < |s| < t < \delta$ and there exists an $s$ such that $\phi(h(y),s) = h \circ \psi(y,t)$, then $s > 0$. In other words, $h$ maps orbits to orbits and preserves the order of time. In this case, we say $\phi$ and $\psi$ are "topologically equivalent".*

3. ***Dynamical equivalence***. *Suppose the flows $\phi$ and $\psi$ have associated vector fields $f$ and $g$. If there exists a homeomorphism $h : \mathbb{R}^2 \to \mathbb{R}^2$ such that*

$$\frac{dy}{dt} = g(y) = f(h^{-1}(y)), \tag{7}$$

*then we say that the flows are "dynamically equivalent."*

If two flows are equivalent according to one of these three notions, then there exists a one-to-one correspondence between their invariant sets which may or may not preserve their stability properties. Note that throughout this paper, we use diffeomorphisms, which is a stronger notion than homeomorphism. In that sense, there are *at least* as many invariances implied in the diffeomorphic case as in the homeomorphic one. In particular, for the notion of "dynamical equivalence", the one we use throughout this paper for its simplicity, we have the following claim:

Claim: Let $X$ and $Y$ be two topological spaces and let $h : X \to Y$ be a homeomorphism, a function that is continuous, bijective, and has a continuous inverse. Let $\dot{x} = f(x)$ be a dynamical system on $X$ and define $\dot{y} = g(y) = f(h^{-1}(y))$. Then, (1) if $f$ has a fixed point, then so does $g$ and (2) if $f$ has a periodic orbit, then so does $g$.

*Proof.* If $f$ has a fixed point at $x_0$ then $f(x_0) = 0$. In that case, $g$ has a fixed point at $y_0 = h(x_0)$ since

$$\begin{aligned} g(y_0) &= f(h^{-1}(y_0)) \\ &= f(x_0) \\ &= 0 \end{aligned}$$

Next, let $x(t)$ be a solution to $f$ which forms a periodic orbit, $\gamma$, passing through $x_0 = x(0)$. $\gamma$ is a simple, closed curve and therefore so is its image, $h(\gamma)$, since $h$ is a homeomorphism. However, we must still show that $h(\gamma)$ is a periodic orbit of $g$. Since $\gamma$ is a periodic orbit of $f$, there exists a $\tau$ such that

$$\begin{aligned} x(\tau) &= x(0) + \int_0^\tau f(x(t)) \, dt \\ &= x(0) \end{aligned}$$

so that $\int_0^\tau f(x(t)) \, dt = 0$. Note that $x(t) = h^{-1}(y(t))$ for all $t$. Thus,

$$\begin{aligned}
y(\tau) &= y(0) + \int_0^\tau g(y(t)) \, dt \\
&= y(0) + \int_0^\tau f(h^{-1}(y(t))) \, dt \\
&= y(0) + \int_0^\tau f(x(t)) \, dt \\
&= y(0),
\end{aligned}$$

which shows that $h(\gamma)$ is a periodic orbit of $g$ with the same period, $\tau$. $\square$

Thus, two ODEs $f(x)$ and $g(y)$ on the plane are "equivalent" ($f \sim g$ via $h^{-1}$) if there exists a homeomorphism $h \in H$ such that $g(y) = f(h^{-1}(y))$ (*). The relation $\sim$ is an equivalence relation partitioning the space of ODEs into dynamical equivalence classes. In particular, this relation is symmetric. For, if * holds, then $g(h(x)) = f(h^{-1}(h(x))) = f(x)$. I.e. $g \sim f$ via $h$.

Since all flows within a dynamical equivalence class share the same features that we care about in this study, if suffices often to work with a single example which has a simple functional form. In that case, we refer to that example as a **prototypical system**.

### 7.1.3 BIFURCATIONS

Next, suppose that the ODEs $\frac{dz}{dt} = f(z; \theta)$ associated to a family of flows is parameterized by $\theta \in \Theta$ which we assume to be one dimensional for simplicity. Then each homeomorphism $h$ partitions the family into dynamical equivalence classes corresponding to a partition of $\Theta$. We have the following definition:

**Definition 5.** *Let $F = \{f(z; \theta) \; : \; \theta \in \Theta \subset \mathbb{R}\}$ be a parameterized family of ODEs corresponding to a family of flows $\phi_\theta$. Suppose $h$ is a homeomorphism defining an equivalence relation $\sim$ on $F$. If there exists a $\theta^*$ such that, for all $\epsilon > 0$, there are $\theta$ such that $\|\theta - \theta^*\| < \epsilon$ with $f_\theta \nsim f_{\theta^*}$, then we say there is a **bifurcation** at $\theta^*$.*

Informally, a bifurcation occurs when smooth changes in a system parameter result in qualitative changes in dynamics. In this study, we focus on classifying vector fields according to whether they belong to one of two equivalence classes in a family of flows exhibiting a supercritical Hopf bifurcation (definition in main text).

## 7.2 METHODOLOGY

### 7.2.1 AUGMENTATIONS

Topological data augmentation was carried out by warping input vector fields with bounded, monotonic rational-quadratic splines proposed in Durkan et al. (2019). The spline mapping $h : x^d \to y^d$ for $d$-dimensional input, where $d$ equals the number of dimensions (in our case, $d = 2$) is based on a set of hyperparameters, including a boundary parameter $B$ and the number of bins $K$. The mapping function $h$ transforms each coordinate within the boundary range $[-B, B]$ using a spline of $K + 1$ knots with each bin in between following a random monotonic rational-quadratic function. Coordinates outside the range retain the identity function (i.e., unchanged coordinates), see Durkan et al. (2019), Figure 1. In our experiments, we set $B$ to 4 and use 5 bins ($K = 5$). Given these hyperparameters, exact parameters defining the splines (width, height and derivatives at the knots) are retrieved from a normalizing flow. Since we use random initialization values, this is equivalent to sampling these parameters directly, without using the normalizing flow architecture. Given a random diffeomorphism, denoted by $h$, and the system dynamics represented by $f$, we invert $h$ to obtain coordinates $X = h^{-1}(Y)$ from the 64-by-64 grid coordinates $Y$. We then apply the dynamics $f$ to the transformed coordinates $X$ and consider these as the dynamics at grid coordinates $Y$, that is $g(Y) = f(X)$. For our augmentations, we adopt an implementation by Andrej Karpathy available at https://github.com/karpathy/pytorch-normalizing-flows.

After applying the augmentation, we verified that the fixed point remained in-frame. This is a proxy to determine if the attractor is still within our field of view. We refer to the dataset obtained by transforming the simple oscillator ("SO") with this augmentation as the "Augmented SO" dataset.

### 7.2.2 ARCHITECTURE AND TRAINING

For vector field $\dot{X} = (\dot{x}, \dot{y})$ of dimensions $[2, 64, 64]$, we first compute its angular representation by replacing each vector with $\arctan{(\dot{y}/\dot{x})}$, which returns a signed representation of the angle in radians.

Our architecture is composed of a convolutional attention feature extraction module based on the SAGAN discriminator (Zhang et al., 2019), coupled with a multi-layered perception (Fig. A5).

The feature extraction module consists of four convolutional blocks, each utilizing a kernel size of $3 \times 3$ and a stride of $2 \times 2$. The number of channels in each layer is progressively increased to 64, 128, 256, and 512. As in SAGAN, we applied spectral normalization and LeakyReLU non-linearities to all convolutional blocks and incorporated the self-attention mechanism into the last two convolutional layers.

The output of the convolutional layers was then fed into a single-layer MLP (multi-layer perceptron) to map the activations to a 10-dimensional space, which we refer to as the "feature activations". The MLP consists of a hidden layer with 64 dimensions, ReLU activation, and a dropout rate of 0.9. Using linear mapping, we predicted the logits for both point and periodic attractors from the feature activations.

We formulated the problem as a multi-class classification task, as there can be simultaneous presence of point and cycle attractors. We used a binary cross-entropy loss.

For training and evaluation, we prepared a dataset comprising 10,000 samples for training and 1,000 samples for testing. We employed the ADAM optimizer with a learning rate of $1 \times 10^{-4}$ and trained the model for 20 epochs, which we confirmed was enough time to fit the training data. For inference, we still perform dropout and average the output of 10 evaluations for each sample.

All experiments were carried out using pytorch v.1.12 using an NVIDIA RTX 2080 GPU, taking $\sim$40 seconds per training experiment.

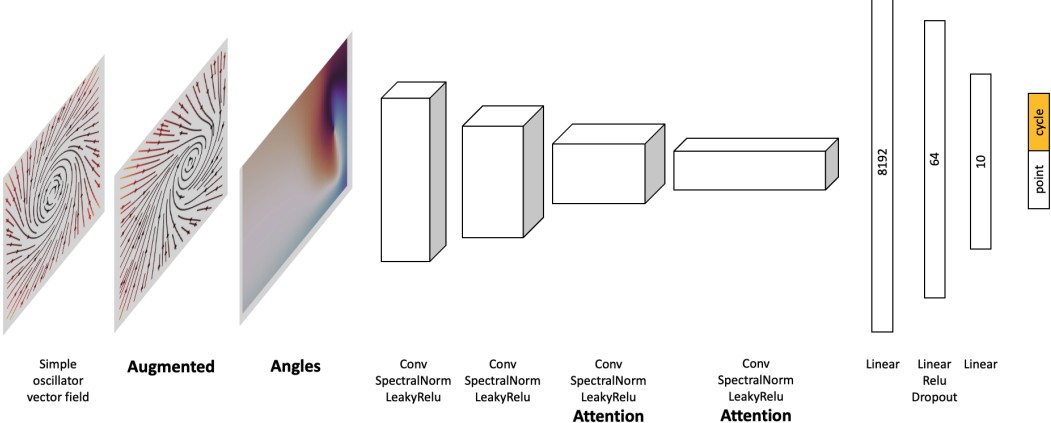

**Figure A5:** *Framework.* Our point-periodic classifier begins by augmenting the vector field data and converting it to an angular representation. Data is then passed through convolutional-attention layers and an MLP classifier.

### 7.3 SYNTHETIC DATASETS

All input vector fields were computed by measuring the continuous system on an evenly-spaced grid of $64 \times 64$ points. In Table A3 the governing equations of each system are described in terms of either polar coordinates $(r, \theta)$ or cartesian coordinates $(x_1, x_2)$ in addition to the range of phase space examined, the parameter ranges we sample from, and bifurcation boundaries described in Strogatz (1994). Examples of pre- and post-Hopf instances are plotted in Fig. A6.

**Table A3:** Descriptions of systems undergoing Hopf bifurcation.
(*) Selḱov periodic attractor condition: $(1/2(1 - 2a - \sqrt{1 - 8a}))^{1/2} < b < (1/2(1 - 2a + \sqrt{1 - 8a}))^{1/2}$

| Name | Equation | Phase space | Parameter ranges | Periodic attractor condition |
|---|---|---|---|---|
| Simple Oscillator | $\dot{r} = r(a - r^2); \dot{\theta} = \omega$ | $x_1, x_2 \in [-1, 1]$ | $a \in [-0.5, 0.5], \omega \in [-1, 1]$ | $a > 0$ |
| Supercritical Hopf | $\dot{r} = \mu r - r^3; \dot{\theta} = \omega + br^2$ | $x_1, x_2 \in [-1, 1]$ | $\mu \in [-1, 1], \omega \in [-1, 1], b \in [-1, 1]$ | $\mu > 0$ |
| Liénard Polynomial | $\dot{x}_1 = x_2; \dot{x}_2 = -(ax_1 + x_1^3) - (c + x_1^2)x_2$ | $x_1, x_2 \in [-4.2, 4.2]$ | $a \in [0, 1], c \in [-1, 1]$ | $c < 0$ |
| Liénard Sigmoid | $\dot{x}_1 = x_2; \dot{x}_2 = -(1/(1 + e^{-ax_1}) - 0.5) - (b + x_1^2)x_2$ | $x_1, x_2 \in [-1.5, 1.5]$ | $a \in [0, 1], b \in [-1, 1]$ | $b < 0$ |
| Van der Pol | $\dot{x}_1 = x_2; \dot{x}_2 = \mu x_2 - x_1 - x_1^2 x_2$ | $x_1, x_2 \in [-3, 3]$ | $\mu \in [-1, 1]$ | $\mu > 0$ |
| BZ Reaction | $\dot{x}_1 = a - x_1 - \frac{4x_1 x_2}{1 + x_1^2}; \dot{x}_2 = bx_1\left(1 - \frac{x_2}{1 + x_1^2}\right)$ | $x_1 \in [0, 10], x_2 \in [0, 20]$ | $a \in [2, 19], b \in [2, 6]$ | $a < \frac{3a}{5} - \frac{25}{a}$ |
| Selḱov | $\dot{x}_1 = x_1 + ax_2 + x_1^2 x_2; \dot{x}_2 = b - ax_2 - x_1^2 x_2$ | $x_1, x_2 \in [0, 3]$ | $a \in [0.01, 0.11], b \in [0.02, 1.2]$ | (*) |

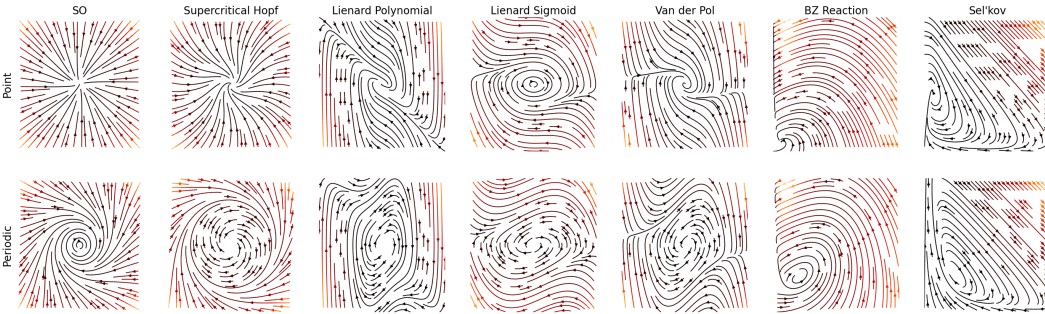

**Figure A6:** *Pre- and post-Hopf examples.* Vector fields visualizations as streamplots of random instances of each system pre- and post-Hopf bifurcation (shown in top and bottom panels respectively).

To assess the parametric proximity of each system to its bifurcation boundary, we calculate the parameter distance as the minimum L2 distance from the bifurcation curve. In the left portion of Fig. A7, we present these signed distances (negative, or positive, distances indicate pre-Hopf, or post-Hopf, systems respectively) for the nonlinear bifurcation curves of the Selḱov system. While parameters fully determine the bifurcation behavior, inferring this behavior from vector fields and trajectories of systems near bifurcations can be misleading. We showcase here instances of the Selḱov systems, each depicted by its vector field and trajectories starting near and away from the fixed point. We can see that for high $b$ values, despite having a point attractor, trajectories exhibit periodic-like behaviors, which can be misleadingly similar to the behavior of a periodic attractor.

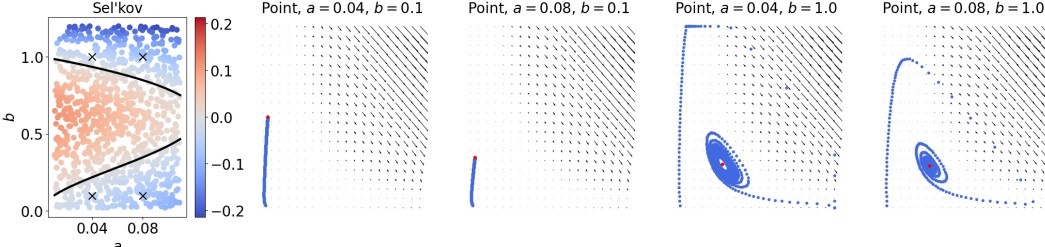

**Figure A7:** *Trajectories of pre-Hopf Selḱov instances.* We examine four instances of the Selḱov system. We show the parameter space (left), its true boundary (black line) and the four samples marked by 'x'. A score (color) is assigned to each system based on how close it is to a bifurcation (minimal distance from the bifurcation curve), and whether it is pre- or post-Hopf bifurcation (of negative or positive scores, respectively). Colors are scaled between the maximum absolute value (red suggesting a cycle) and its negation (blue) with white at zero value. On the right, we show vector fields with a flow $F(x_0, T)$ where $x_0$ is away (blue), or near (red) the fixed point. We set $T = 100$ and interpolation is over timesteps of 0.1.

We tested our proposed method on noisy data which was subjected to additive zero-centered Gaussian noise ($\mu = 0$) and scaling $\sigma = 0.1$ (see Fig. A8 and Table 1) and on increasing scaling of $\sigma$ for Augmented SO (see Table 2).

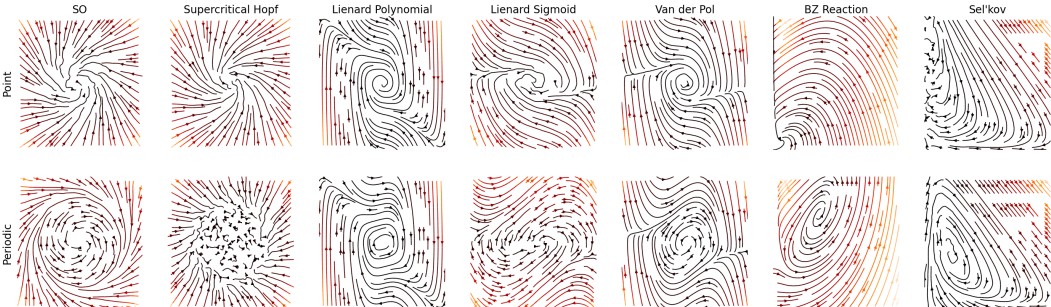

**Figure A8:** *Perturbed pre- and post-Hopf examples.* Gaussian noise of $\sigma = 0.1$ added to vector fields of pre- and post-Hopf examples (shown in top and bottom panels respectively) of each system.

## 7.4 BASELINES

The advantage of learning a topology-aware representation is emphasized by comparison to other across-system representations, namely: (1) "Critical Points", (2) "Lyapunov", (3) "Phase2vec", (4) "Autoencoder", and (5) "Parameters".

**Critical Points.** Since we focus here on Hopf bifurcations, a point and a periodic attractor can be distinguished by the identity of the fixed point, being an attractor for the former and a repeller for the latter. Helman & Hesselink (1989; 1991) devised an algorithm for detecting and characterizing such critical points from an autoencoder representation, later implemented in `https://github.com/zaman13/Vector-Field-Topolgy-2D`. Running this method in practice, we find that for a point attractor, a single point is often detected but for a periodic attractor, the algorithm mistakenly outputs multiple coordinates as fixed points besides its repeller (mean number of points for point attractor is 1.00 and for periodic attractor 1.50). We therefore adjust the rule to classify a system as a periodic attractor if some identified critical point is of repelling type. Using this measure, we find that the classification of the synthetic systems is nearly perfect (see Table A4), but breaks immediately with noise; with Gaussian noise of $\sigma = 0.1$ added to the Augmented SO dataset, accuracy drops to 59%.

**Lyapunov.** Given time-series data, the maximal Lyapunov exponent, $\lambda_1$, captures the greatest separation of infinitesimally close trajectories across an n-dimensional space. Systems with $\lambda_1 < 0$ are considered in a state of fixed point, whereas periodic states have $\lambda = 0$ (Stoker, 1950). Zhang et al. (2009) uses this measure in the context of Hopf bifurcations towards the design of rotational machines. To evaluate this measure, we integrate each system from a random point at an interval of 0.1 up to time $T = 100$ based on the vector field and compute the maximal Lyapunov exponent, $\lambda_1$, of the trajectory using the python library `https://pypi.org/project/nolds`. We find the optimal threshold of $\lambda_1$ for classification based on Augmented SO ROC curve and use it across all systems (see Table A4).

**Phase2vec.** For Phase2vec we follow the tutorial of data generation and training as described in Ricci et al. (2022) and implemented in `https://github.com/nitzanlab/phase2vec`. That is, we generate 10K train samples of sparse polynomial equations and train the phase2vec framework consisting of a convolutional encoder mapped by a single linear layer into a $d = 100$-dimensional embedding space and consequently decoded via another linear layer into predicted parameter values which are multiplied by the polynomial library to reconstruct the vector fields. We then train a linear classifier of pre-Hopf and post-Hopf classes using the same Augmented SO dataset as used in our current framework and select for optimized hyperparameters (see Table A4).

**Autoencoder.** For Autoencoder we deploy an autoencoder architecture consisting of an encoder of three convolutional blocks (kernel size of $3 \times 3$, a stride of $2 \times 2$, and 128 channels in each layer), followed by a linear layer into 10 latent dimensions, and a decoder mirroring the encoder architecture. The autoencoder is trained on the same data used in phase2vec with ADAM optimizer with a learning rate of $1 \times 10^{-4}$, for 20 epochs. We then train a linear classifier of pre-Hopf and post-Hopf classes using the same Augmented SO dataset as used in our current framework and select for optimized hyperparameters (see Table A4).

**Parameters.** For Parameters, a vector of 20 coefficients, $(c_0, c_1, ..., c_9, d_0, d_1, ..., d_9)$, is fitted using least squares to reconstruct the two-dimensional vector field $(\dot{x}, \dot{y})$ with $(q_3(x, y|\mathbf{c}), q_3(x, y|\mathbf{d}))$ where $q_3(x, y|\mathbf{c}) = c_0 + c_1 x + c_2 y + c_3 x^2 + c_4 y^2 + c_5 xy + c_6 x^3 + c_7 y^3 + c_8 x^2 y + c_9 xy^2$. We chose polynomial representation of degree 3 following earlier work (Iben & Wagner (2021)) which showed this value was sufficient for dynamical systems similar to those we investigate here. We then train a linear classifier of pre-Hopf and post-Hopf classes using the same Augmented SO dataset as used in our current framework and select for optimized hyperparameters (see Table A4).

**Table A4:** Test accuracy of baselines and their hyperparameter tuning, default: learning rate=1e-4, num epochs = 20.

| | SO | Augmented SO | Supercritical Hopf | Lienard Poly | Lienard Sigmoid | Van der Pol | BZ Reaction | Sel'kov | Average |
|---|---|---|---|---|---|---|---|---|---|
| Our Model | 0.97±0.01 | 0.93±0.01 | 0.98±0.01 | 0.85±0.14 | 0.92±0.1 | 0.84±0.14 | 0.84±0.09 | 0.65±0.03 | 0.87 |
| learning rate = 1e-3 | 0.95±0.09 | 0.91±0.09 | 0.96±0.09 | 0.84±0.17 | 0.95±0.09 | 0.89±0.1 | 0.73±0.12 | 0.65±0.07 | 0.86 |
| learning rate = 1e-5 | 0.97±0.01 | 0.86±0.02 | 0.98±0.01 | 0.88±0.1 | 0.79±0.13 | 0.76±0.14 | 0.53±0.09 | 0.53±0.07 | 0.79 |
| num epochs = 10 | 0.97±0.01 | 0.92±0.01 | 0.98±0.01 | 0.88±0.12 | 0.86±0.13 | 0.83±0.15 | 0.74±0.13 | 0.63±0.04 | 0.85 |
| num epochs = 40 | 0.98±0.01 | 0.93±0.01 | 0.98±0.01 | 0.86±0.13 | 0.93±0.06 | 0.85±0.14 | 0.81±0.11 | 0.67±0.06 | 0.88 |
| Critical Points | 1.00 | 0.93 | 1.00 | 1.00 | 1.00 | 1.00 | 0.70 | 0.92 | 0.94 |
| Lyapunov | 0.85 | 0.71 | 0.89 | 0.89 | 0.87 | 0.84 | 0.78 | 0.68 | 0.81 |
| Phase2vec | 0.89±0.06 | 0.86±0.02 | 0.91±0.02 | 0.51±0.12 | 0.48±0.0 | 0.5±0.05 | 0.5±0.0 | 0.49±0.0 | 0.64 |
| learning rate = 1e-3 | 0.9±0.05 | 0.87±0.03 | 0.93±0.02 | 0.61±0.16 | 0.7±0.14 | 0.76±0.17 | 0.51±0.05 | 0.49±0.02 | 0.72 |
| learning rate = 1e-5 | 0.85±0.03 | 0.73±0.01 | 0.94±0.01 | 0.47±0.0 | 0.56±0.07 | 0.48±0.0 | 0.51±0.04 | 0.49±0.0 | 0.63 |
| num epochs = 10 | 0.81±0.06 | 0.82±0.02 | 0.91±0.02 | 0.47±0.0 | 0.48±0.0 | 0.48±0.0 | 0.5±0.0 | 0.49±0.0 | 0.62 |
| num epochs = 40 | 0.92±0.04 | 0.87±0.02 | 0.9±0.01 | 0.49±0.07 | 0.5±0.05 | 0.5±0.06 | 0.5±0.0 | 0.49±0.0 | 0.65 |
| Autoencoder | 0.93±0.03 | 0.86±0.01 | 0.96±0.01 | 0.65±0.18 | 0.92±0.08 | 0.85±0.11 | 0.83±0.13 | 0.5±0.02 | 0.81 |
| learning rate = 1e-3 | 0.94±0.03 | 0.92±0.01 | 0.96±0.01 | 0.63±0.15 | 0.96±0.04 | 0.8±0.17 | 0.64±0.15 | 0.64±0.1 | 0.81 |
| learning rate = 1e-5 | 0.93±0.04 | 0.65±0.03 | 0.96±0.01 | 0.47±0.02 | 0.8±0.16 | 0.54±0.11 | 0.53±0.11 | 0.52±0.09 | 0.67 |
| num epochs = 10 | 0.9±0.03 | 0.82±0.02 | 0.95±0.01 | 0.52±0.12 | 0.88±0.12 | 0.6±0.17 | 0.71±0.14 | 0.49±0.0 | 0.73 |
| num epochs = 40 | 0.95±0.02 | 0.9±0.01 | 0.96±0.01 | 0.53±0.01 | 0.92±0.08 | 0.7±0.13 | 0.69±0.13 | 0.56±0.09 | 0.78 |
| Parameters | 0.53±0.05 | 0.52±0.03 | 0.56±0.06 | 0.52±0.11 | 0.53±0.1 | 0.54±0.12 | 0.53±0.1 | 0.52±0.07 | 0.53 |
| learning rate = 1e-3 | 0.5±0.0 | 0.57±0.01 | 0.49±0.01 | 0.47±0.0 | 0.48±0.0 | 0.48±0.0 | 0.5±0.0 | 0.49±0.0 | 0.5 |
| learning rate = 1e-5 | 0.5±0.08 | 0.5±0.03 | 0.5±0.13 | 0.51±0.12 | 0.49±0.18 | 0.5±0.13 | 0.51±0.12 | 0.51±0.08 | 0.5 |
| num epochs = 10 | 0.53±0.07 | 0.5±0.03 | 0.54±0.07 | 0.51±0.12 | 0.51±0.2 | 0.53±0.15 | 0.49±0.07 | 0.5±0.04 | 0.51 |
| num epochs = 40 | 0.54±0.07 | 0.56±0.02 | 0.57±0.08 | 0.52±0.1 | 0.55±0.14 | 0.55±0.13 | 0.54±0.11 | 0.5±0.05 | 0.54 |

## 7.5 ABLATIONS OF OUR MODEL

To evaluate the impact of our architecture, we compare the results with ablated versions of our full model, namely: (1) "No Attention" - where we exclude the two self-attention layers, resulting in a network that is purely feedforward, (2) "From Vectors" - our model but trained on raw vector fields instead of transformed angles as inputs, i.e. with two channels in the first convolutional layer, (3) "No Augmentations" - our model but trained on the unaugmented SO data, and (4) "CNN-baseline" - a model including all ablations, see Table A5.

**Table A5:** Test accuracy on ablated models.

| | SO | Augmented SO | Supercritical Hopf | Liénard Poly | Liénard Sigmoid | Van der Pol | BZ Reaction | Sel'kov | Average |
|---|---|---|---|---|---|---|---|---|---|
| Our Model | 0.97±0.01 | **0.93±0.01** | 0.98±0.01 | 0.85±0.14 | 0.92±0.1 | 0.84±0.14 | **0.84±0.09** | **0.65±0.03** | **0.87** |
| No Attention | 0.97±0.01 | 0.92±0.01 | 0.98±0.01 | 0.63±0.13 | 0.66±0.12 | 0.53±0.07 | **0.84±0.09** | 0.62±0.05 | 0.77 |
| From Vectors | 0.97±0.02 | 0.92±0.01 | 0.98±0.01 | 0.66±0.12 | **0.96±0.03** | **0.93±0.05** | 0.62±0.15 | 0.57±0.05 | 0.82 |
| No Augmentation | **0.99±0.0** | 0.45±0.01 | **0.99±0.0** | **0.94±0.0** | **0.96±0.0** | 0.88±0.01 | 0.5±0.01 | 0.52±0.05 | 0.78 |
| CNN-baseline | 0.98±0.01 | 0.55±0.01 | **0.99±0.01** | 0.59±0.13 | 0.77±0.14 | 0.61±0.1 | 0.45±0.09 | 0.49±0.0 | 0.68 |

We observe good performance when training on classical SO for systems that resemble the SO in having their fixed point located persistently at the center. In fact, as we increase the boundary parameter $B$ of the neural spline augmentations, which directly affects the location distribution of the fixed point, we observe the change in test accuracy for two systems: the BZ reaction system, which exhibits a peripheral fixed point (with $x_1$ ranging between 0.6 and 3.8 within the $[0, 10]$ limits) ; and the Liénard Polynomial system, which has a centrally located fixed point, see Fig. A9 and Table A6.

Furthermore, we examine the rules learned by the attention layers. We sum the values of the first attention mask and expand it to the original grid dimensions. Through this analysis, we observe that the attention is often sparse, focusing on the fixed point or its immediate neighbors, as depicted in Fig. A10. This observation raises intriguing follow-up questions like whether higher accuracy correlates with specific patterns in the attention masks.

**Data size and resolution** We assayed the average accuracy as a function of data size, showing that a substantially smaller train data size (2k instead of 10k) is sufficient in most cases, see Table A7. In the case of a higher sampling resolution in space, our method is directly applicable. Indeed, for

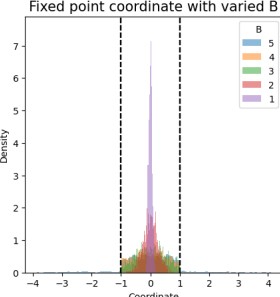

| B | 1 | 2 | 3 | 4 | 5 |
|---|---|---|---|---|---|
| BZ Reaction | 0.42 | 0.65 | 0.79 | 0.88 | 0.71 |
| Liénard Poly | 0.92 | 0.90 | 0.86 | 0.79 | 0.89 |

**Figure A9 & Table A6:** Effect of $B$, the boundary parameter in Neural Spline Flows, on the fixed point coordinate distribution (see Figure, dashed black lines mark the train data boundaries) and on test accuracy for BZ Reaction and Liénard Polynomial (see Table).

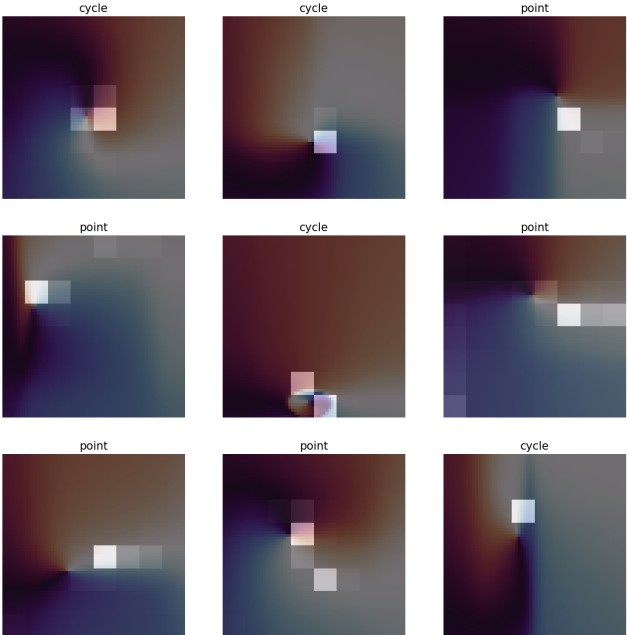

**Figure A10:** *Attention learned.* Mask of the first attention layer is diluted back to the original grid dimensions for a random (but constant) point shown in grayscale (standardized by minimum and maximum values in black and white respectively).

the current resolution (64x64), training was quick, at $\sim 40\text{s}$, while training at twice the resolution resulted in a training time of $\sim 130\text{s}$ and the same test accuracy (87% average accuracy).

**Table A7:** Average test accuracy with different train data sizes.

| Train data size (k) | 1 | 2 | 4 | 8 | 16 |
|---|---|---|---|---|---|
| Average accuracy | 0.7 | 0.84 | 0.85 | 0.87 | 0.89 |

**Other architectural choices.** We observe robust performance upon editing our network architecture, see Table A8.

**Table A8:** Test accuracy of alternative representations varying neural network architecture parameters.

| | SO | Augmented SO | Supercritical Hopf | Lienard Poly | Lienard Sigmoid | Van der Pol | BZ Reaction | Sel'kov | Average |
|---|---|---|---|---|---|---|---|---|---|
| Our Model | 0.97±0.01 | 0.93±0.01 | 0.98±0.01 | 0.85±0.14 | 0.92±0.1 | 0.84±0.14 | 0.84±0.09 | 0.65±0.03 | 0.87 |
| dropout = 0.5 | 0.98±0.01 | 0.93±0.01 | 0.98±0.01 | 0.84±0.17 | 0.93±0.06 | 0.8±0.14 | 0.76±0.12 | 0.68±0.06 | 0.86 |
| kernel size = 5 | 0.97±0.01 | 0.92±0.01 | 0.98±0.01 | 0.93±0.05 | 0.93±0.03 | 0.87±0.07 | 0.72±0.11 | 0.6±0.05 | 0.87 |
| latent dimension = 5 | 0.98±0.01 | 0.93±0.01 | 0.99±0.01 | 0.88±0.12 | 0.93±0.08 | 0.85±0.12 | 0.79±0.11 | 0.65±0.04 | 0.87 |
| latent dimension = 20 | 0.97±0.01 | 0.93±0.01 | 0.98±0.01 | 0.84±0.15 | 0.91±0.09 | 0.87±0.08 | 0.82±0.11 | 0.65±0.05 | 0.87 |
| conv channels = 32 | 0.98±0.01 | 0.92±0.01 | 0.99±0.01 | 0.86±0.13 | 0.91±0.08 | 0.87±0.09 | 0.76±0.11 | 0.67±0.06 | 0.87 |
| conv layers = 3 | 0.97±0.01 | 0.92±0.01 | 0.98±0.01 | 0.88±0.12 | 0.92±0.1 | 0.86±0.14 | 0.79±0.11 | 0.66±0.04 | 0.87 |

## 7.6 RECOVERING BIFURCATION BOUNDARIES

In Fig. A11 we plot for each system our model's confidence as a function of its parametric proximity to its nearest bifurcation point. In Fig. A12 we depict the bifurcation diagrams for the alternative trained baselines.

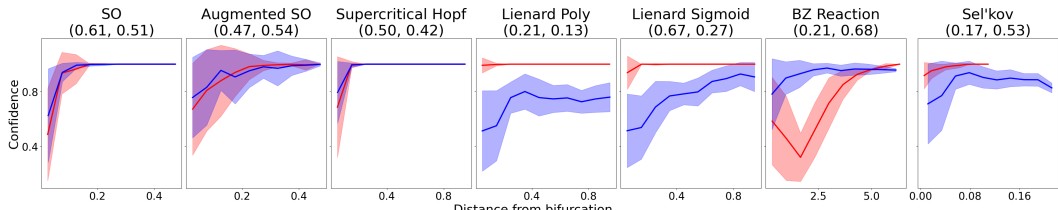

**Figure A11:** *Low confidence near bifurcation boundaries.* For each system, we plot mean (line) and standard deviation (shaded region) of the confidence over the 50 training runs as a function of the parametric distance from the bifurcation boundary for pre-Hopf (blue) and post-Hopf (red) systems with the respective Spearman's correlations noted in the title.

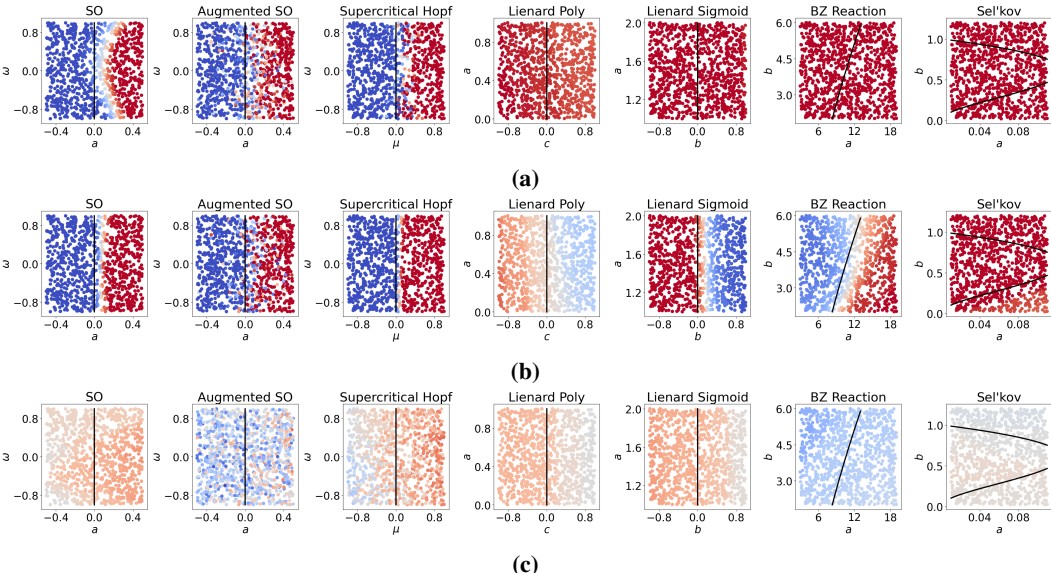

**Figure A12:** *Inference of bifurcation boundary for alternative representations.* As in Fig. 2, each system is plotted at coordinates of its simulated parameter values and colored according to its average cycle prediction across 50 training re-runs based on *(a)* Phase2vec, *(b)* Autoencoder, and *(c)* Parameters. The true parametric bifurcation boundary is plotted in black. Colors are scaled between a perfect cycle classification (red) and perfect point classification (blue) with white at evenly split classification.

## 7.7 CO-OCCURRENCE OF ATTRACTORS

There are systems where both point and periodic attractors are present, for example in Subcritical Hopf bifurcation (Strogatz, 1994):

$$\dot{r} = \mu r + r^3 - r^5,$$
$$\dot{\theta} = \omega + br^2$$

with a point attractor where $\mu < -0.25$, a point and a periodic attractor where $-0.25 < \mu < 0$, and a periodic attractor where $\mu > 0$. The average (across runs) point logit our framework outputs for each regime is, respectively: 1.32, -6.72, -11.03.

## 7.8 REPRESSILATOR SYSTEM

Gene regulation of the repressilator system is formulated as a set of six equations, or three sets of the following two equation (Elowitz & Leibler, 2000):

$$\dot{m}_i = -m_i + \frac{\alpha}{1 + p_j^n} + \alpha_0,$$
$$\dot{p}_i = -\beta(p_i - m_i)$$

where $m_i$, or $p_i$ describe the mRNA, or protein concentration, of gene $i$ which is one of LacI, TetR and $\lambda$ phage cI genes. The mRNA expression of gene $i$ is inhibited by the protein levels of gene $j$ for the corresponding gene pairs of $i$ = LacI, TetR, cI and $j$ = cI, LacI, TetR. $\alpha_0$ is the rate of leaky mRNA expression (that is, transcription taking place despite being entirely repressed), $\alpha$ is transcription rate without repression, $\beta$ is the ratio of protein and mRNA degradation rates, and $n$ is the Hill coefficient.

We set the parameters $\alpha_0 = 0.2, n = 2$, and consider trajectories of cells of varying transcription rate $\alpha \in (0, 30)$, and of varying ratio of mRNA and protein degradation $\beta \in (0, 10)$. A supercritical Hopf bifurcation occurs across both boundaries,

$$\beta_1 = \frac{3A^2 - 4A - 8}{4A + 8} + \frac{A\sqrt{9A^2 - 24A - 48}}{4A + 8},$$
$$\beta_2 = \frac{3A^2 - 4A - 8}{4A + 8} - \frac{A\sqrt{9A^2 - 24A - 48}}{4A + 8}$$

where $A = \frac{-\alpha n \hat{p}^{(n-1)}}{(1+\hat{p}^n)^2}$ and $\hat{p} = \frac{\alpha}{1+\hat{p}^n} + \alpha_0$, (Verdugo, 2018).

We plot these boundaries in Fig. 3 and Fig. A13. The discontinuity between $\beta_1$ and $\beta_2$ in the plots is due to our discretized computation of the boundary; we take a grid of $\alpha$ values and solve for their upper ($\beta_1$) and lower ($\beta_2$) boundaries.

To generate a vector field we simulate a trajectory of 50 timesteps sampled at 0.1 intervals starting at $(m_{LacI}, p_{LacI}, m_{TetR}, p_{TetR}, m_{cI}, p_{cI}) = (2.11, 2.28, 1.57, 1.71, 1.07, 1.14)$, see examples in Fig. A13. We sample 100 cells from this trajectory, add Gaussian noise of $\sigma = 0.5$ and evaluate the six-dimensional vectors of these cell states. In Fig. A14a we show examples of these cell vectors as they are projected onto the TetR-LacI protein plane together with their theoretical pre-Hopf or post-Hopf categorization. We then interpolate these vectors to a 64×64 grid (Fig. A14b), and follow-up with angle transformation (Fig. A14c).

## 7.9 PANCREAS DATASET

We obtain the data, compute high-dimensional gene expression velocities per cell and their projection into UMAP space, and each cell's cell-cycle score as described in scVelo (Bergen et al., 2020), RNA velocity basics tutorial. Since cell types present distinct dynamics (Fig. A15), in order to enrich our dataset, we partition cells into batches of at least 50 cells of common cell type (Fig. A16a). We then generate samples for our classifier by interpolating their sparse two-dimensional velocities to a 64-by-64 grid (Fig. A16b), which are later transformed into angle representations (Fig. A16c). From

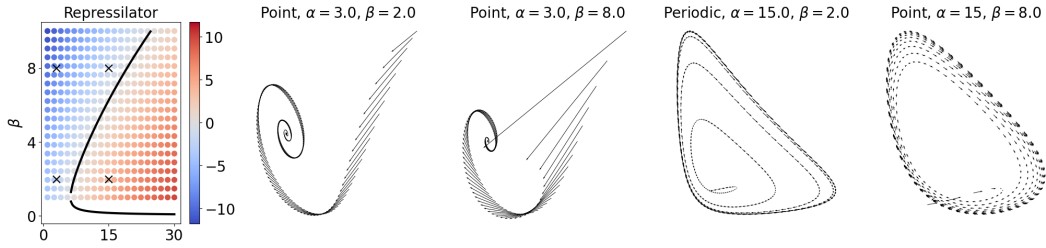

**Figure A13:** *Examples of repressilator trajectories.* We examine four cellular trajectories of the repressilator system. We show the parameter space (left), its true boundary (black line) and the four samples marked as 'x'. A score (color) is assigned to each system based on how close it is to a bifurcation (minimal distance from the bifurcation curve), and whether it is pre- or post-Hopf bifurcation (of negative or positive scores, respectively). Colors are scaled between the maximum absolute value (red suggesting a cycle) and its negation (blue) with white at zero value. On the right, we show complete cellular trajectories with $T = 50$ and intervals of $0.1$.

these, we predict using our classifier whether cells demonstrate cyclic dynamics, where in this case, such dynamics are characteristic of the cell cycle. To test our evaluation, we aggregate individual cell cycle scores into a population score as the fraction of cells that are undergoing S-phase, where a score threshold of 0.4 is employed (Fig. A16d).

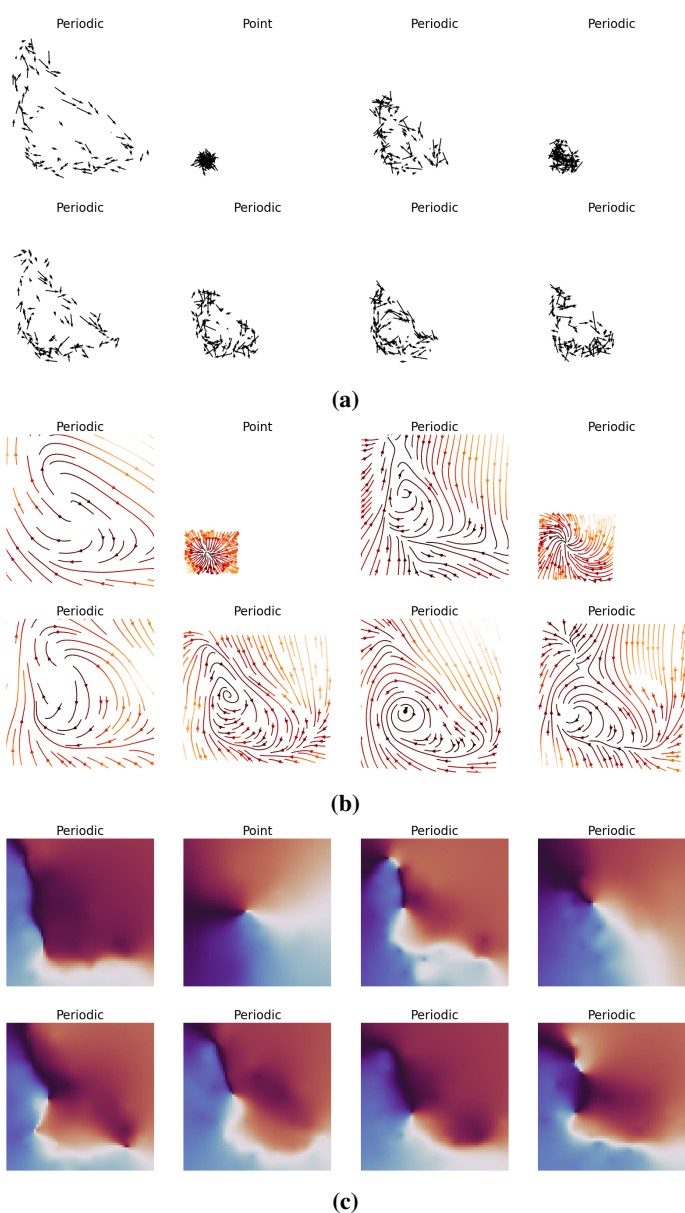

**Figure A14:** *Repressilator simulated samples.* We show here the data generation process for the repressilator system of eight random instances (distinct $\alpha, \beta$ values). We simulate noised trajectories of 100 cells and project their velocities onto the TetR and LacI protein plane *(a)*. From these, we interpolate vector values to a 64-by-64 grid *(b)* and image their corresponding angles *(c)*. Titles indicate the theoretical pre- and post-Hopf classification of each sample.

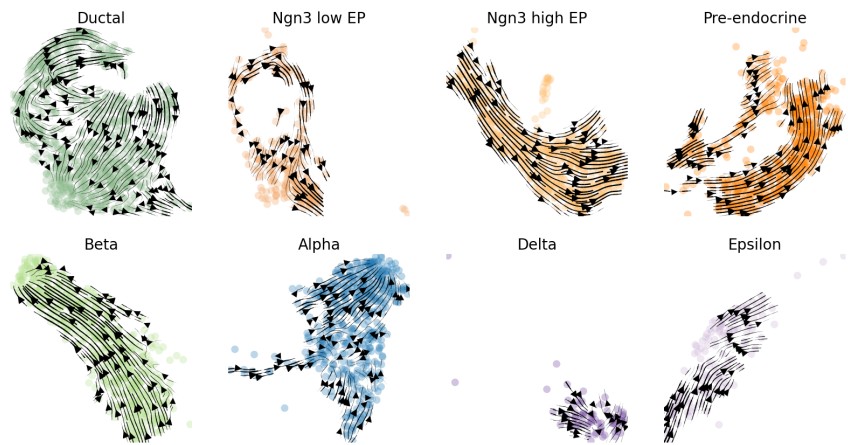

**Figure A15:** *Cell type velocities.* Depicting the velocities in UMAP representation for each cell type separately.

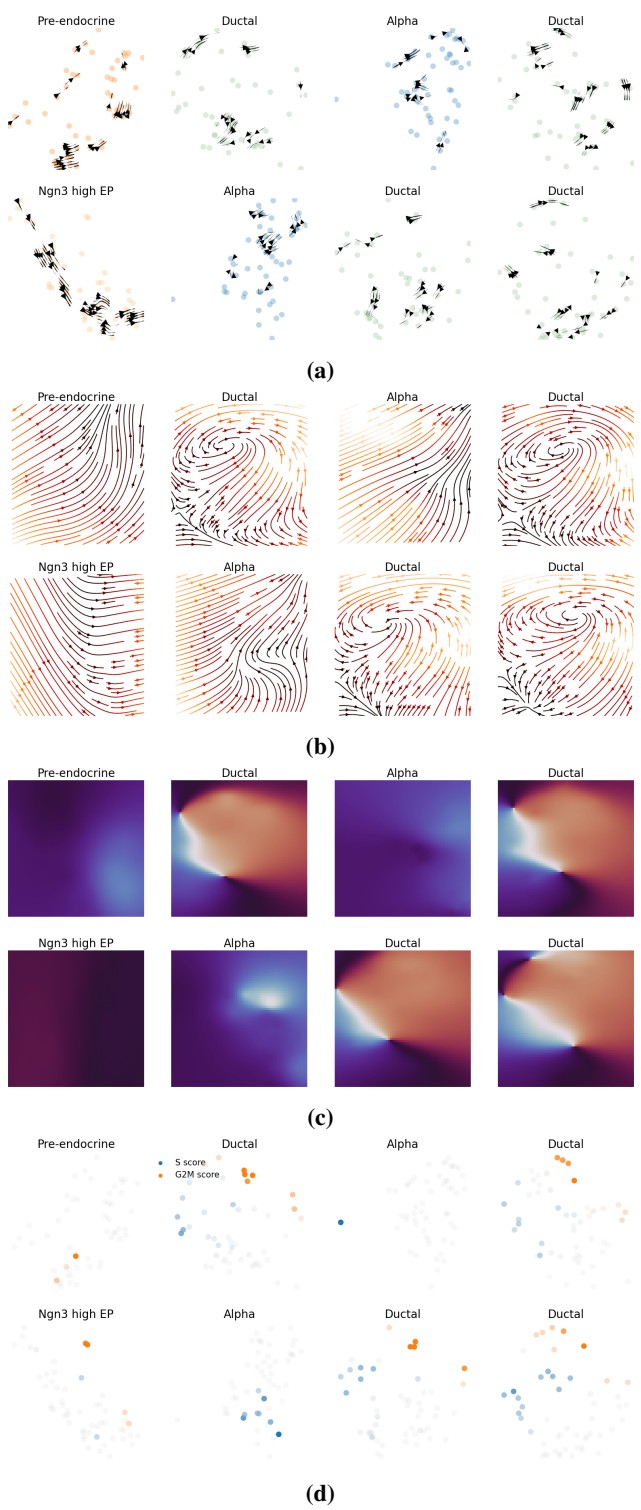

**Figure A16:** *Pancreas data samples.* We randomly split cells of common cell type into groups of 50 or more cells. Based on their velocities *(a)*, we compute interpolated vectors *(b)* and their angles *(c)*. We compute a proxy for the cell-cycle signal in each sample, which we consider as the fraction of cells undergoing S-phase according to their S-score value *(d)*.

