# OpenReview forum: "Let's do the time-warp-attend: Learning topological invariants of dynamical systems"
_ICLR.cc/2024/Conference — ICLR 2024 poster_

### Official Review · Reviewer_sMRU · 2023-10-20

**Soundness:** 3 good
**Presentation:** 2 fair
**Contribution:** 4 excellent
**Rating:** 6
**Confidence:** 4

**Summary:**

In this paper, the authors present a deep-learning framework designed for the classification of dynamical regimes and the characterization of bifurcation boundaries. Their approach is rooted in the extraction of topologically invariant features. The effectiveness of this method is convincingly demonstrated through its application to real-world data. Notably, the framework excels in the analysis of such data, successfully delineating distinct proliferation and differentiation dynamics along the pancreatic endocrinogenesis trajectory within the gene expression space.

**Strengths:**

- The paper introduces a novel approach to topologically invariant feature learning in dynamical systems, utilizing warped vector field data as augmentations to generalize across systems with similar dynamics in their topological representations within phase space. The method seems novel to me.
- The paper demonstrates the effectiveness of the proposed method through its application to real data, successfully recovering distinct proliferation and differentiation dynamics within the gene expression space of the pancreatic endocrinogenesis trajectory. This suggests a broad range of applications across various fields, including biology, physics, and beyond.
- The paper holds promise for making a substantial impact in the field of machine learning for dynamical systems, offering a new approach to addressing the challenge of predicting impending catastrophes in diverse real-world systems.
- The paper also suggests promising future research directions, including expanding the scope of invariance types and enhancing equivalence notions.

**Weaknesses:**

- The methodology section appears disorganized. The addition of more subtitles would greatly enhance readability. Moreover, the frequent transition between the main content and the appendix disrupts the reading experience.
- The model setup details being located in the appendix can make it less accessible, as it's separated from the primary content.
- While the wording and sentences in the experiments section are well-constructed, there's room for improvement in the organization of tables and figures. Consider enlarging figures and addressing font size issues, particularly in tables. For instance, Table 3's column headers could be better aligned with their respective columns.
- The process for reproducing the results remains unclear. It would be beneficial to include a dedicated "Reproducibility Statement" section, providing readers with the necessary guidance to replicate the findings.

**Questions:**

In addition to the points mentioned in the 'Weaknesses' section, several questions arise:

- Regarding the model setup, it would be valuable to understand why the current configuration is deemed the most suitable for this problem. An ablation study, exploring architectural variations such as layers with different sizes, would provide insights into the model's design choices.
- Are there specific hyperparameters within this method that require tuning, and if so, what is the recommended approach for hyperparameter optimization?
- A dedicated 'Limitations' section would be a useful addition to provide a comprehensive perspective on the constraints and challenges of the proposed approach."

---

> ### Author Response · Authors · 2023-11-22
>
> We thank the reviewer for the helpful and practical comments. We have adjusted the manuscript accordingly, specifically:
>
> 1. We have clarified the text and divided the Methodology section into subjects. To improve clarity and flow, we moved some of the model details from the Appendix to the Methodology and kept the more technical details in the Appendix. We framed and centered tables’ content and adjusted font size where plausible.
>
> 2. We now provide an anonymous link to our preliminary repository as a footnote at the end of the Introduction section.
>
> 3. We added results for varying hyperparameters (learning rate and training epochs) and architectural choices (dropout rate, kernel size, MLP latent dimension size, number of convolutional channels, and number of convolutional layers), see Tables A4 and A8. We found no significant effect on performance for these hyperparameters and ablations, except for a reduced performance with learning rate = 1e-5.
>
> 4. We added a dedicated limitations section, touching upon the challenges of combinatorial annotations and scaling to higher-dimensional systems.

---

> > ### Comment · Reviewer_sMRU · 2023-11-22
> > **Thanks for the rebuttal and revision.**
> >
> > Dear Authors,
> >
> > Thank you for your response to the comments. The adjustments you've made to the manuscript, particularly the reorganization and clarification of the Methodology section, greatly enhance the paper's clarity and flow. The decision to move certain model details from the Appendix into the main body of the paper, while keeping the more technical aspects in the Appendix, is a wise one that balances detail with readability. I am happy with its current form.
> >
> > Warm regards,
> > Reviewer sMRU

---

> > > ### Author Response · Authors · 2023-11-22
> > >
> > > We agree, the problem, method, and results are communicated much better now after integrating the reviewer’s suggestions. We are grateful for the reviewer’s feedback.

---

### Official Review · Reviewer_YdjJ · 2023-10-28

**Soundness:** 2 fair
**Presentation:** 2 fair
**Contribution:** 1 poor
**Rating:** 3
**Confidence:** 4

**Summary:**

Authors propose a data-driven, physically-informed deep-learning framework for classifying dynamical regimes and characterizing bifurcation boundaries based on the extraction of topologically invariant features. Authors further demonstrate the method's use in analyzing real data, recovering distinct proliferation and differentiation dynamics along pancreatic endocrinogenesis trajectory in gene expression space based on single-cell data.

**Strengths:**

1. The experiments are extensive.
2. The related works are well organized.

**Weaknesses:**

1. The paper is written poorly and hard to read. I cannot find the problem definition. What's the meaning of prototypical system?
2. The main method is based on convolutional attention method and data augmentation, which is widely used in dynamical system modeling.
3. The methodology part needs to be reorganized. Authors should divide that into different parts, which can be related to the contributions. Now, it only shows data augmentation and an existing network to me.
4. More SOTA methods should be compared. For example, there are extensive RNA velocity methods [1].

[1] Deep generative modeling of transcriptional dynamics for RNA velocity analysis in single cells, 2023.

**Questions:**

See above.

---

> ### Author Response · Authors · 2023-11-22
>
> We thank the reviewer for their remarks.
>
> Consequently, we have extensively revised the introduction and methods to clarify our problem setting. In brief, our aim is to learn a limit cycle detector which is robust both to nuisance features (cycle shape, frequency, etc) and noise. We achieve this with the topological data augmentation method described in the manuscript, which ensures that those nuisances are ignored in favor of the underlying topological structure which defines the bifurcation classes. Our system is trained on augmented versions of a simple system (the “prototype system”, in this case the simple oscillator system whose equations can be found in “Dynamical equivalence and topological augmentation” subpart in Methodology) which concisely captures the main features of these classes. We have adjusted the text to emphasize these details.
>
> Further, in the Related Works and in the Results sections, we now argue and demonstrate that existing topological methods are either very sensitive to noise, or require system-specific prior knowledge, like the explicit form of the governing equation or the true bifurcation parameter.
>
> To our knowledge, current data-driven methods for bifurcation detection deal largely with the problem of predicting the onset of a bifurcation from time series data measured from a single system. Our work is the first to tackle a much more general problem: a data-driven approach to automatically delineate dynamical classes across a huge number of real and synthetic systems varying both in their functional and parametric form. And, while the reviewer is correct that convolutional architectures are not uncommon in data-driven dynamical systems, they are certainly rare in this novel problem setting.
>
> Our paper makes several key contributions in service of this goal. First, it introduces a novel approach for learning topologically invariant features (further clarified in the revised text) in dynamical systems by leveraging warped vector field data. Second, it achieves invariant classification of complex synthetic data. Third, the paper demonstrates the use of classifier confidence to effectively identify bifurcation boundaries. Last, the proposed method is successfully applied to analyze biological systems, specifically uncovering distinct dynamics in the pancreatic differentiation trajectory using single-cell gene expression data where alternative baselines fail (see updated Results section 4.4).
>
> Last, veloVI (Gayoso et al. 2023), the method the reviewer mentioned, is a generative framework of RNA dynamics. It is an alternative to the upstream steps prior to our method. In our analysis we use the current convention in the field, applying standard preprocessing steps (e.g. filtering genes, normalizing expression, etc.) and extraction of velocities with scVelo. For specifically analyzing single-cell RNA velocity data, the coupling of veloVI and our framework in the future can be very interesting. It would require introducing confidence across phase space but will also better mitigate its partial coverage.

---

> ### Comment · Reviewer_YdjJ · 2023-11-23
>
> Thanks for your response. Now the paper is till challenging for many machine learning researchers to understand due to the use of specific technical terms like "cycle detector" and "prototype system." To enhance clarity and accessibility, it would be beneficial for the authors to provide definitions for each technical term they use. This approach can make the paper more comprehensible to a broader audience, including those who may not be familiar with these specific concepts.

---

> ### Author Response · Authors · 2023-11-23
>
> Given the interdisciplinary nature of our framework, which spans dynamical systems, topology, and machine learning, we aimed to offer comprehensive background information and diverse references to cater to a broad readership. We believe that our manuscript and the works it cites provide adequate background for a machine-learning audience. Indeed, we believe that the manuscript is also much clearer than the initial draft, as has been recognized by other reviewers. In the final draft, we will reproduce standard mathematical definitions in a glossary in the appendix.

---

### Official Review · Reviewer_YHJc · 2023-10-29

**Soundness:** 3 good
**Presentation:** 3 good
**Contribution:** 3 good
**Rating:** 6
**Confidence:** 5

**Summary:**

This paper introduces a method to identify the existence of fixed points and limit cycles in dynamical systems based on their vector fields in order to able to detect supercritical Hopf bifurcation boundaries. This method relies on a convolutional feed-forward network with attention and data augmentation that relies on warping. This method is then applied to some toy systems and single-cell data to identify different regimes in parameter space and differentiation transitions respectively.

**Strengths:**

This is to our knowledge the first method to perform topological invariant feature learning using deep learning.
The method has a high performance to detect bifurcation boundaries in various dynamical systems and even on real world data. Especially the application to detection of cell cycle score is a surprising contribution.
Furthermore, the effectiveness of the approach is demonstrated against a couple of other methods highlighting its applicability.
The paper is well written and present with a good flow to explain clearly the contributions. The particular contributions of the different parts of the network architecture and training method provides useful insights.  Robustness to relatively high levels of noise seem to further suggest the usefulness of this method for real world applications.
Identifying bifurcations is an important problem in dynamical systems theory and this paper shows some promising results to tackle it.

**Weaknesses:**

\paragraph{Minor comments}
"In the left portion of Fig. ??" should have been "In the left portion of Fig. A7"?

The reference "ML Cartwbight. Balthazar van der Pol, 1960." seems erroneous.

\paragraph{Performance on systems that have both features}
It would greatly benefit the demonstration of the usefulness of the method, if there would be an assessment of the performance on systems that have both a fixed point and a limit cycle.



\paragraph{Figure 2}
It is surprising that the correlation coefficient for the BZ reaction is higher than Supercritical Hopf, while the boundary seems quite off.

\paragraph{Use of different training set sizes for the different methods}
The comparison of the method proposed in the paper to phase2vec and the Vector Field Learned Representation seem unfair considering that the models have been trained on a different number of training samples. Why is your model trained on 10K training samples why the others are trained on only 7.5K. This could explain why phase2vec is performing worse.

It would further clarify the differences between the methods if you could furthermore show how the inference of the bifurcation boundaries look like for the other methods.

Finally, how do the other methods perform with the addition of noise?

\paragraph{Use of the chosen hyperparameters}
The particular choice for the used hyperparameters is insufficient and makes the comparison to the other methods less convincing. See also the Questions. Showing the dependence of the performance of the methods on the different parameters would show how sensitive the method is to hyperparameter tuning.
Or stating that the best performing hyperparameters were chosen for each method would make the comparison better.


% Significance:
\paragraph{Comparison  to other methods}
To fully assess the contribution of this work a more extensive set of method to compare to should be considered.
First, of all, Lyapunov exponents could be used to track bifurcations and the existence of limit cycles (Witte, 2014). This method work for any dimension in principle, not just 2.

Furthermore, the Conley-Morse graph (Arai, 2009) (has more information, works in a more general setting (other dimensions than 2)) should be considered.
Is this method faster? More accurate than constructing the Conley-Morse graph or the method proposed in the paper?

Finally, one could track down bifurcations through continuation algorithms (Veltz, 2020), see
\url{https://docs.juliahub.com/BifurcationKit/I1INQ/0.1.4/detectionBifurcation/}.


The performance of these other reliable methods would give a better idea of the usefulness of the proposed method.



De Witte, V., Govaerts, W., Kuznetsov, Y.A. and Meijer, H.G.E., 2014. Analysis of bifurcations of limit cycles with Lyapunov exponents and numerical normal forms. Physica D: Nonlinear Phenomena, 269, pp.126-141.

Veltz, R., 2020. BifurcationKit. jl (Doctoral dissertation, Inria Sophia-Antipolis).

**Questions:**

Why was a learning rate of $10^{-4}$ chosen for training? Why 20 epochs? How was it "confirmed" that that "was enough to fit the training data"? Why were 20 coefficients used for the Fitted Parameter Coefficients?


What is the discontinuity of the theoretical boundary in Figure 3 and Figure A12 relating to?

How are cells partitioned into batches of at least 50 cells? How is the size furthermore determined for each batch? Does the partitioning have any consequences for the resulting model?

---

> ### Author Response · Authors · 2023-11-22
>
> We appreciate the important and constructive comments raised by the reviewer. We have made the following adjustments to the manuscript:
>
> 1. Minor comments - These are now corrected in the manuscript.
>
> 2. We added a new experiment in the appendix regarding systems that have mixed limit cycle / fixed point behavior. Note that we use a supervised, multi-class setup so that the classifier can predict both the existence of a limit cycle and a fixed point. Empirically, we find that our method learns that the point and periodic classes are exclusive. For emphasis, we tested our trained model on samples of a subcritical Hopf bifurcation (see Appendix 7.6 having regimes of point attractor, point and periodic attractors, and periodic attractor). We find that, even though our system has never seen this bifurcation, it can detect the monotonic relationship between class score and the order parameter. In the future, we aim to extend to combinatorial annotations of invariant sets. We report this as a limitation and in the discussion converse about how we intend to address this in follow-up work.
>
> 3. Figure 2 - Pearson correlation assesses linear relationships.  Since the confidence of the BZ reaction oscillator gradually increases with the distance from the bifurcation, its Pearson correlation is very high, as opposed to systems like Supercritical Hopf whose confidence quickly plateaus. We now plot the confidence as a function of the distance from the bifurcation boundary in Figure A11 alongside Spearman’s correlation to quantify the monotonic relationship.
>
> 4. Use of different training set sizes for the different methods - We adopted the reviewer’s suggestion of training baselines on train data of equal size of 10K. This had little effect on the performance of the baselines (see Table A4).
>
> 5. Effect of noise and bifurcation boundary diagrams across methods - In Tables 1 and 2 we now benchmark the performance of baselines with noise and their bifurcation boundary diagrams in Figure A12.
>
> 6. Use of the chosen hyperparameters - In Table A4 we report test accuracies for Phase2vec, Autoencoder and Parameters across learning rates and epoch iterations. Some hyperparameters improved the average accuracy, e.g. for Phase2vec, changing the learning rate from 1e-4 to 1e-3 increased accuracy from 66% to 72%, however, the ranking of methods across systems is maintained. For all learning rates except 1e-5, our method achieved consistently high accuracies, even when varying architectural parameters, see Table A8.
>
> 7. Comparison to other methods - To motivate our approach to learning and generalization, we have added new numerical experiments which compare to methods making use of data topology, namely, via (1) heuristics based on critical points, and (2) Lyapunov exponent thresholding, see Appendix 7.3 for details and Tables A4 and 1 for performance on noiseless and noisy data. We observe that the Lyapunov exponent method is more sensitive to noise than our approach (see Table 2). The Conley Morse Graph Database proposed in (Arai et al. 2009) assumes knowledge of the governing equations, so we did not consider it. The Conlety-Morse graph computation from vector representation developed in (Chen et al. 2007; Chen et al. 2008) was not accessible as the software only works in Windows 10. Finally, numerical continuation algorithms operate by iteratively constructing intersecting solution curves in conjoined phase-parameter space. In most practical settings, we do not have access to parameter space and must contend with a dataset of isolated measurements, for example, time series or vector fields. We do not know how those measurements relate to one another in parameter space, and we certainly lack the ability to parametrically interpolate between those samples in a continuous way. The practical data scientific question then becomes ”how does one detect invariant sets in individual examples?”, which is what our paper strives to address.
>
> 8. As we now note in the revised manuscript, we chose polynomial representation of degree 3 following the earlier work of (Iben and Wagner 2021) which showed this value was sufficient for dynamical systems similar to those we investigate here.
>
> 9. The discontinuity of the theoretical boundary of the repressilator system is due to our discretized computation of the boundary. We take a grid of alpha values and solve for their upper and lower boundaries. We mention this now in Appendix 7.7.
>
> 10. For constructing the pancreas dataset, we first partition cells according to their cell type. Given n cells and minimal size s=50, we generate k=floor(n/s) labels, bin the cells into k buckets and then permute the labels (see notebooks/pancreas.ipynb in our repository) . Having observed the 2D grids that emerge with the current batch size (see Appendix Fig A13) we were satisfied since we observed a clear distinction between the dynamic regimes and were able to generate a sizable cohort of 69 samples.

---

> > ### Comment · Reviewer_YHJc · 2023-11-23
> >
> > Thank you for the response. I appreciate the additional details and the new version is taking steps in the right direction.
> >
> > The implementation of computing the Conley-Morse graph should be quite straightforward and would still give a very general approach that is not bounded to two dimensions (although calculations can  become very time-consuming in higher dimensions). There exists also software that to my awareness is not only for Windows 10: https://github.com/shaunharker/CHomP.
> >
> > As a fair comparison to state-of-the field tools is lacking, I am maintaining my score.

---

> > > ### Author Response · Authors · 2023-11-23
> > >
> > > We appreciate the reviewer’s diligence in going through our manuscript and providing insightful references and constructive feedback. We are actively engaged in the process of incorporating a benchmark involving the computation of the Conley-Morse graph, as suggested, to further enhance the robustness and completeness of our work.

---

### Official Review · Reviewer_pimK · 2023-10-31

**Soundness:** 2 fair
**Presentation:** 3 good
**Contribution:** 2 fair
**Rating:** 6
**Confidence:** 3

**Summary:**

The paper proposes a method for classifying dynamical regimes in a data driven deep learning framework, combining a self-attention GAN for feature extraction and training an MLP for classification.
Their main goal is to extract invariant topological properties from dynamical systems around bifurcations, and use them to classify dynamical regimes. To this end they leverage a data augmentation method, based on the principle of topological/dynamical equivalence. The first pre-train their model on variants of a simple oscillator system featuring a supercritical Hopf bifurcations.
They apply their pre-trained model to classify two regimes (pre-Hopf and post-Hopf) in several two-dimensional, autonomous systems (both synthetic and experimental data), featuring the same bifurcation. To this end they show that they can extend knowledge of dynamical classes (distinguishing between fixed point and cycles) across different datasets, and determine bifurcation boundaries and closeness to bifurcations.

**Strengths:**

The ideas in the paper are clearly presented. The idea of detecting bifurcations in dynamical systems is timely and important for applications. Their framework for data augmentation, based on the principle of dynamical/topological equivalence, is interesting, and the idea of transfer learning across different dynamical systems that share common topological properties seems like an important application. I also appreciate the application to experimental cell data, and the ablation studies in the supplement, investigating the effect of data augmentation and different modules, e.g. the attention modules, on outcomes.

**Weaknesses:**

Related Work/Dynamical Systems Theory:

One of my main concerns is that there is little reference to the rich mathematical body of work in dynamical systems theory anywhere in the paper. The concept of "dynamical equivalence" proposed in the supplement is well known in the literature (see e.g. Kutznetsov, "Elements of Applied Bifurcation Theory), as are the proofs given in 6.1. in the context of the concepts of topological conjugacy&topological equivalence. However, both terms are only mentioned once in passing in the whole paper, and as far as I could see are not referenced anywhere. The term "topological invariants", while used in the abstract and title, is also never formally introduced nor explained in the rest of the paper.

My second main concern is that I don't understand your reasoning why you "evaluate as baselines existing vector field representations, none of which explicitly encourage topological invariance". Topological features (such as fixed points, cycles, stability of fixed points etc.) can also be numerically approximated directly from the flow fields, especially for simple 2-D models you investigate here where the vector field is already present (i.e. fixed points are simply locations where the flow field is zero). There are other topological properties (e.g. persistent homologies, Betti numbers) that one could also extract to aid with classification.  At least using this as a baseline for comparison would seem crucial in the context of this paper? If I'm not completely mistaken this should at least work better than random guessing, as the comparison methods on the Selkov model effectively do. Fitting polynomial/other representations to the vector fields that a priori are not tailored to extract a useful representation for topological invariants seems perhaps unsurprisingly not so effective?
It could also be interesting in this context to investigate to what extent what your model extracts can be intepreted, and how it relates to other topological invariants.

Restriction to 2D

The restriction of all experiments to 2-D is also quite limiting. The authors indeed mention "technical challenges" in their conclusion for extensions to higher dimensions, but there are deeper theoretical reasons for these challenges, e.g. the lack of structural stability in higher dimensions, and more generally the scaling of your approach with the dimensionality of the dynamical system. Since this is a substantial limiting factor for extending your method to experimental settings, this could be made more explicit in the discussion.


In summary, in case I did not fundamentally misunderstand something about your approach and the general problem setting, it is unclear to me why you would need such a sophisticated machine learning architecture to extract features that can in the situation you investigate could be estimated from the flow directly? Since this is a crucial point to my mind point I am voting for rejection at this point, but am willing to adjust my score if these concerns are adressed.

**Questions:**

Your examples indicate that augmentation actually decreases performance for the simpler baseline models, which goes a little counter to the argument of your paper. Why do you think this is the case?

---

> ### Author Response · Authors · 2023-11-22
>
> We thank the reviewer for the valuable insights and constructive feedback. We have made significant revisions to the manuscript to extend the discussion on related works and underscore the motivation behind our approach over and above existing methods which make use of data topology. As we now demonstrate in several new numerical experiments, these earlier methods fare significantly worse in classifying noisy vector field data, both synthetic and real, as they are either tricked by spurious signals from fixed points or struggle to learn a single decision boundary across all systems. In contrast, our method learns features which are both robust and universal across systems, making our method more viable in practical settings than previous techniques.
>
> We address the main points raised by the reviewer:
>
> 1. Following the reviewer’s suggestion, we have extended the Methodology section to better contextualize our method in the dynamical systems literature. In particular, in the subsection  “Topological equivalence in dynamical systems”, we discuss the literature cited by the reviewer (Kutznetsov and the standard notions of topological conjugacy and equivalence) and precisely what we mean by the term topological “invariant”. In the subsequent subsection, “Dynamical equivalence and topological augmentation”, we elaborate on our augmentation scheme in light of this literature. We are now particularly careful to distinguish the term “dynamically equivalent” from similar notions in the literature while highlighting its role in preserving limit cycles and fixed points, which is what we prove in Appendix 7.1.1.
>
> 2. We thank the reviewer for recommending new baselines to justify our approach. We emphasize first that our aim is eventually to classify real data, and our reasoning is that generalization based on learned neural features will be more robust to the noisy conditions inherent to real data than existing measures based on direct, hardwired measurements of flow. For example, the reviewer suggested using a heuristic based on fixed points might obviate the need for learning altogether. We implemented such a heuristic and found that it achieves nearly perfect classification accuracy for noiseless synthetic systems but entirely fails when even a little noise is added ( see Appendix 7.3 for method details and Tables A4 and 1 for performance). Similarly, classification accuracy using this heuristic for the single-cell pancreas dataset is only 54%. Persistent homology, as well as other dynamical measures like Lyapunov exponents (Stoker 1992; Zhang et al. 2009), rely on time-series data, which is not always available, e.g., in single-cell RNA velocity data. We have added a benchmark that generates trajectories from the vector field and computes its Lyapunov exponent. This method, too, was more sensitive to noise than our approach (see Table 2). Concerning persistent homology, we currently followed (Berwald et al. 2013) in using the two longest bars of degree 1 persistence barcode for classification, but observed low accuracy on the train data (63%), hence, we did not include it in the attached submission but are currently pursuing its integration. We are highly motivated to further explore the rules learned by the model, to compare these to existing theory, and to leverage such understanding for extensions of the framework.
>
> 3. In a new section on the limitations of our approach, we address the restriction to two dimensions along with other constraints. We would first note that preprocessing by dimensionality reduction can mitigate difficulties from high dimensions, as we use for the analysis of the six-dimensional repressilator, or for the single-cell data with over 2K dimensions. What’s more, low-dimensional dynamics are still an active area of study, as our experiments in single-cell trajectories show. Indeed, superficially high-dimensional dynamics in biology are often intrinsically low-dimensional and suited to our approach (see for instance Xiong and Garfinkel 2023). Exponential scaling with dimensions is inherent to the vector field format, although we imagine several ways to circumvent this problem. Some recent works have used graph neural networks or transformers to learn compact representations of phase space from sparse data. We will examine these extensions in future work.
>
> 4. We did not entirely follow the reviewer’s observation: "Your examples indicate that augmentation actually decreases performance for the simpler baseline models, which goes a little counter to the argument of your paper." In general, the augmentations mitigate the overfitting to simple systems, as discussed in Appendix 7.4. Although we train on Augmented SO data, performance on unseen, simpler test systems, like SO, achieve higher accuracy across frameworks, as opposed to the trained Augmented SO system, because the augmented data is very complex, being asymmetric, off-center, and sometimes with limit cycles partially out of frame.

---

> > ### Comment · Reviewer_pimK · 2023-11-22
> >
> > Thanks for the detailed feedback and for adressing many of my concerns. I will recheck the paper and novel experiments and am happy to improve my score accordingly, also in light of further discussions with the other reviewers.
> >
> > A quick clarification: regarding the 4th point, I was refering the results in Table A5, where for the SO, Supercritical Hopf and the Lienard systems, performance without augmentation outperformed that of the full method.
> > Regarding lower-D dynamics in 3), this is also commonly observed in neuroscience, where low-D latent embeddings are often sought, and dynamical properties thought to be involved in cognitive mechanisms, so this could also be mentioned in the discussion/related work.

---

> > > ### Author Response · Authors · 2023-11-22
> > >
> > > We thank the reviewer for the quick response and we look forward to continued feedback. To address your points in turn:
> > >
> > > There are several shared properties among the Supercritical Hopf, Lienard, Van der Pol and unaugmented SO system which we believe explain the observed effect. Notably, in all these cases, both point and periodic attractors have a fixed point in the center of the phase space grid, as illustrated in Figure A6. Without augmentation, test and train (SO) data are similar. With augmentations, on the other hand, training data is more complex, with the all-important fixed point appearing at varied locations. This makes the training data and the Supercritical Hopf, Lienard, and Van der Pol systems quite different, inhibiting generalization. We discuss this in Appendix 7.4 and plan to investigate other explanatory factors (e.g., symmetry, size of limit cycle, etc.) on our model’s generalization performance in the future.
> > >
> > > The reviewer’s remark about low-D dynamics specifically in neuroscience is well-taken. We are reminded of the famous toroidal structure of neural activity in grid cell populations (Gardner et al. 2022) as well as the ubiquity of low-dimensional attractor dynamics (Khona and Fiete 2022). We are also aware of recent work at the whole-brain scale showing that BOLD dynamics live on low-dimensional manifolds during free viewing of videos (Song et al. 2023). We will integrate this material into the introduction/discussion section along with discussion about how dimensionality reduction methods can be appended to our approach in a single pipeline.

---

### Author Response · Authors · 2023-11-22

We thank the reviewers for their constructive comments. In response to these remarks, we have carried out a number of follow-up experiments, detailed here, and have adjusted the manuscript for clarity and to emphasize the novelty of our approach. We believe that both our problem setting and the method we use to address it are novel. The problem we consider here is the data-driven, generalizable classification of a diverse set of both real and synthetic dynamical systems in realistic noise conditions into two important topological equivalence classes. This broad scope fundamentally distinguishes our work from existing methods in bifurcation analysis which deal exclusively with the prediction of oncoming catastrophes from time series data of single, isolated systems. The fact that our method can be simultaneously applied to numerous systems of diverse functional and parametric forms – without retraining – demonstrates that our model has learned an abstract notion of these equivalence classes. Further supporting our approach is a new series of benchmarks detailed below.

We have adjusted the manuscript scientifically and terminologically. On the scientific front, we have carried out a series of new experiments inspired by the reviewers’ insightful remarks:

1. We have benchmarked our topologically-informed method against established topological measures based on critical points or on the Lyapunov exponent, see Tables A4 and 1. For the former, classification rules of critical points achieve nearly perfect accuracy on synthetic systems but break easily with noise. For the latter, we first note that like most existing methods for differentiating periodic and aperiodic systems, the Lyaponuv exponent requires time-series data as opposed to the vector field representation in our problem setting. Further, Lyapunov exponent analysis also results in a substantial decrease in accuracy, e.g. for the Supercritical Hopf system. We discuss how these methods could be extended or integrated with our own in the Discussion section.

2. We have evaluated the robustness of our method as a function of architectural (number of convolutional layers, number of convolutional channels, the MLP latent dimension, and dropout rate, see Table A8) and optimization hyperparameters (learning rate, training epochs, see Table A4), obtaining almost consistently an overall accuracy of between 0.85 and 0.88. These supplement our previous robustness analysis over train dataset size and vector field resolution, see Appendix 7.4.

We also rigorized our notation and problem scope in the main text and in a much extended and organized Methods section. In particular, we have clarified our exact problem setting (distinguishing samples having fixed points from those with limit cycles for the case of the supercritical Hopf bifurcation) and explained our use of particular warpings. These warpings preserve the underlying topology of the two classes under consideration, which we clarify both here and in the manuscript. Moreover, when training data is augmented with these warped samples, we see a marked improvement in performance over our newly included baselines.

Crucially, we have also noted, here and in the revised manuscript, the limitations of our method and extended the discussion section. First, though not an inherent limitation, we chose to focus on the minimal, two-class example of the supercritical bifurcation. Extending to the multi-class case or the two-class subcritical case is an area of future work. Second, although we would like our representations to be invariant to geometric (as opposed to topological) properties of systems, these nuisance features nevertheless play a role in the final classification decision, as manifest in the confidence of this decision. As a result, despite overall high accuracy in classification, bifurcation boundaries are affected by the confidence values. Next, our method is currently limited to the important but circumscribed setting of low-dimensional dynamics. We discuss potential approaches for scaling to the higher-dimensional setting here and in the revised manuscript.

Recognizing the strengths and limitations inherent in our methodology, we believe that our refined manuscript significantly enhances the accessibility of topological analysis for vector fields, particularly in the context of real-world data, such as applications in biology, clinical and climate settings.

---

### Meta-Review · Area_Chair_bFyz · 2023-12-05

**Metareview:**

This paper introduces a novel method for detecting topological invariants of dynamical systems using a data-driven deep learning approach. The reviewers found the results in detecting bifurcation boundaries strong, and generally agree that the paper tackles an important problem, is well-written, and achieves impressive results.

**Justification For Why Not Higher Score:**

The paper could be better grounded in the literature, with a more direct comparison to existing methods. The broader applicability and impact of the results is perhaps rather limited.

**Justification For Why Not Lower Score:**

This paper addresses a problem that is relatively unexplored in the prior literature, namely performing topological invariant feature learning using deep learning. The good results suggest the possibility of opening up a new direction that many researchers may pursue in the future.

---

### Decision · Program_Chairs · 2024-01-16

Accept (poster)